# ATF4 leads to glaucoma by promoting protein synthesis and ER client protein load

Ramesh B. Kasetti[1], Pinkal D. Patel[1], Prabhavathi Maddineni[1], Shruti Patil [1], Charles Kiehlbauch[1],
J. Cameron Millar[1], Charles C. Searby[2], VijayKrishna Raghunathan [3,4], Val C. Sheffield [2] & Gulab S. Zode[1✉]

The underlying pathological mechanisms of glaucomatous trabecular meshwork (TM) damage and elevation of intraocular pressure (IOP) are poorly understood. Here, we report that the chronic endoplasmic reticulum (ER) stress-induced ATF4-CHOP-GADD34 pathway is activated in TM of human and mouse glaucoma. Expression of ATF4 in TM promotes aberrant protein synthesis and ER client protein load, leading to TM dysfunction and cell death. These events lead to IOP elevation and glaucomatous neurodegeneration. ATF4 interacts with CHOP and this interaction is essential for IOP elevation. Notably, genetic depletion or pharmacological inhibition of ATF4-CHOP-GADD34 pathway prevents TM cell death and rescues mouse models of glaucoma by reducing protein synthesis and ER client protein load in TM cells. Importantly, glaucomatous TM cells exhibit significantly increased protein synthesis along with induction of ATF4-CHOP-GADD34 pathway. These studies indicate a pathological role of ATF4-CHOP-GADD34 pathway in glaucoma and provide a possible treatment for glaucoma by targeting this pathway.

[1] Department of Pharmacology and Neuroscience and the North Texas Eye Research Institute, University of North Texas Health Science Center at Fort Worth, TX 76107 Fort Worth, USA. [2] Department of Pediatrics, Carver College of Medicine,  University of Iowa, Iowa City, IA 52242, USA. [3] Department of Basic Sciences and the Ocular Surface Institute, College of Optometry,  University of Houston, Houston, TX, USA. [4] Department of Biomedical Engineering, Cullen College of Engineering,  University of Houston, Houston, TX, USA. ✉email: gulab.zode@unthsc.edu

G laucoma is a heterogeneous group of optic neuropathies characterized by progressive loss of retinal ganglion cells (RGCs), degeneration of the optic nerve, and progressive visual field loss[1–3]. Glaucoma is the second leading cause of irreversible blindness, affecting ~70 million people worldwide[4]. In the United States, primary open-angle glaucoma (POAG) is the most common form of this disease and it is especially prevalent in African-American and Hispanic populations[5–9]. Elevated IOP is the most important and the only known clinically modifiable risk factor for the development and progression of POAG[10]. IOP is tightly regulated by aqueous humor production by the ciliary body and its outflow through trabecular meshwork (TM) tissue. In POAG, there is increased resistance to aqueous humor outflow at the TM, elevating IOP[2,11,12]. This increased outflow resistance is associated with several biochemical and morphological changes in the TM[13]. Glaucomatous TM damage includes increased synthesis and deposition of extracellular matrix (ECM), reduced phagocytosis, formation of cross-linked actin networks and loss of TM cellularity[13–19]. However, the pathological mechanisms responsible for increased outflow resistance at the TM are poorly understood.

A variety of glaucomatous insults including mutant myocilin (MYOC) and glucocorticoid (GC) therapy are known to damage TM, elevating IOP[20,21]. MYOC mutations are responsible for ~4% of POAG and most cases of autosomal dominant juvenile-onset-open-angle glaucoma[22,23]. Ocular hypertension is a serious side effect of GC therapy. Approximately 30–40% of the general population develop elevated IOP as a result of topical ocular administration of potent GCs such as dexamethasone (Dex) or betamethasone. If undetected, GC-induced ocular hypertension can lead to secondary open-angle glaucoma that mimics POAG[24–28]. Similar to POAG, GC-induced glaucoma is caused by increased resistance to aqueous humor outflow[21,28–30], as well as morphological and biochemical changes in the TM including increased ECM accumulation[31–34].

Although POAG is a complex disease and caused by several factors, study of these factors has led to discovery of common pathogenic mechanisms. Recent studies have revealed the role of chronic endoplasmic reticulum (ER) stress in the pathophysiology of POAG[35–40]. Specifically, expression of mutant MYOC or Dex treatment leads to chronic ER stress in the TM[37–39,41]. The ER is a central organelle for synthesis, folding, and maturation of secretory and membrane proteins. ER stress develops when the ER is overwhelmed by unfolded and misfolded proteins, exceeding its handling capacity. In eukaryotes, ER stress is sensed by a constellation of three signal transduction pathways, collectively known as the unfolded protein response (UPR), which aid in the restoration of ER homeostasis and thus normal function[42,43]. The UPR can have either protective or destructive effects on cellular functions, which in turn depend on the strength and/or duration of ER stress. Chronic and sustained UPR generally causes cell death[44,45]. One such mechanism of chronic ER stress-mediated cell death involves activation of PERK, which leads to phosphorylation of the eukaryotic translation initiation factor (eIF2α) resulting into translational block. Activating transcription factor 4 (ATF4) is preferentially translated, which further activates downstream targets including C/EBP homologous protein (CHOP)[46]. ATF4 and CHOP directly induce the growth arrest and DNA damage-inducible protein (GADD34), which further promotes dephosphorylation of eIF2α leading to protein translation recovery[47]. There is an emerging theme that IRE1-XBP-1 activation occurs transiently in ER stress and is cell protective, whereas PERK-ATF4-CHOP activation persists during chronic ER stress and triggers cell death[48–50]. Although deletion of CHOP prevents ER stress-mediated cell death, recent studies have suggested an essential role for ATF4 in

chronic ER stress-mediated cell death[51,52]. Importantly, it is not completely understood how chronic ER stress factors lead to cell death.

Although our previous studies have established that chronic ER stress is associated with TM dysfunction and IOP elevation, it is not understood whether and how chronic ER stress leads to TM dysfunction and IOP elevation. Moreover, it is not understood which signaling factors in the ER stress pathway play pathological roles in IOP elevation. Since ATF4 and CHOP are known to play pathological roles in several diseases associated with ER stress, and our recent finding that ATF4 and CHOP are significantly induced in post-mortem TM tissues from POAG donor eyes[36], we further explored the roles of ATF4 and CHOP in the glaucomatous TM damage and IOP elevation.

## Results

### Induction of ATF4–CHOP–GADD34 pathway in human and mouse glaucoma.
Our previous studies were focused on induction of CHOP in mouse models and human glaucoma donor eyes[36,38,39]. In the present study, we further examined whether ATF4–CHOP–GADD34 signaling axis is also activated in TM cells and tissues of human and mouse glaucoma. We first examined markers of chronic ER stress in human primary TM cells obtained from normal and glaucoma donor eyes ($n = 3$-6 different cell strains). Western blot (Fig. 1a) and its densitometric analyses (Fig. 1b) demonstrated significantly increased ATF4, CHOP, GADD34, and spliced XBP-1 protein levels in glaucomatous primary TM cells. Immunostaining for ATF4 and GADD34 in the TM tissues of human donor eyes obtained from age-matched normal ($n = 9$) and POAG patients ($n = 11$) further revealed that both ATF4 and GADD34 are significantly increased in POAG eyes (Fig. 1c, d and Supplementary Fig. 1). We next examined whether activation of ATF4–CHOP–GADD34 signaling axis is associated with IOP elevation using the recently developed mouse model of Dex-induced ocular hypertension[53]. C57BL/6J mice were treated weekly with periocular injections of vehicle or Dex for several weeks, and night IOPs were monitored weekly[53]. Dex treatment led to significant and sustained IOP elevation starting at 1-week of injection (Fig. 1e). Dex treatment led to increased IOP of 4.9 mmHg at week 1, 6.2 mmHg at week 2, 12.1 mmHg at week 3, 10.4 mmHg at week 4, and 8.5 mmHg at week 5 of injections. Western blot and its densitometric analyses (Fig. 1f, g) demonstrated that GRP78, ATF4, and CHOP protein levels are significantly increased in Dex-treated iridocorneal angle tissues. Spliced XBP1 and GADD34 levels were also increased in Dex-treated eyes, although this increase was not found statistically significant. In addition, immunostaining for ATF4 along with alpha smooth muscle actin (αSMA) revealed that ATF4 levels are prominently increased in the TM tissues of 5-week Dex-treated mice (Supplementary Fig. 2). Increased αSMA was also observed in the TM tissues of Dex-treated mice. Consistent with our previous studies, we observed increased ER stress (KDEL; recognizes GRP78 and GRP94) in TM of Dex-treated mice (Supplementary Fig. 3), and increased ATF4 and CHOP in GTM3 cells treated with Dex (Supplementary Fig. 4). Both ATF4 and CHOP were mostly localized to nuclei suggesting transcriptional activity. These studies indicate that ATF4–CHOP–GADD34 signaling pathway is induced in the glaucomatous TM and that induction of these proteins is associated with IOP elevation.

### ATF4 elevates IOP and reduces aqueous humor outflow in mice.
We next examined whether adenoviral expression of ATF4 elevates IOP in C57BL/6J mice. Three-month-old mice were injected intravitreally with Ad5.ATF4 or Ad5.Empty ($2 \times 10^7$ pfu/eye), and conscious IOP was measured once per week for up to

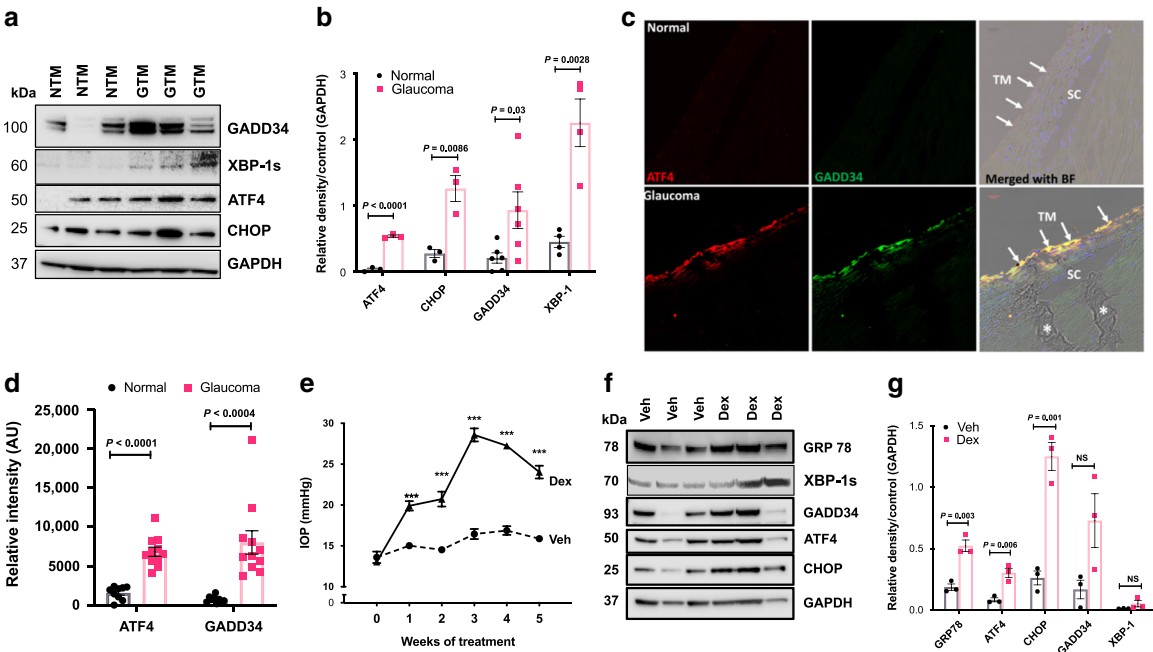

**Fig. 1 ATF4–CHOP–GADD34 is induced in TM cells/tissues of human and mouse glaucoma. a** and **b** Cellular lysates from age-matched normal and glaucomatous primary human TM cells were examined for ER stress markers using Western blotting **a** and analyzed by densitometric analysis **b**. The average from two independent experiments is shown graphically ($n = 3$ cell strains for ATF4 and CHOP; $n = 6$ cell strains for GADD34; $n = 4$ cell strains for XBP-1; data are presented as mean ± SEM, 2-tailed unpaired $t$-test). **c** and **d** Representative immunostaining for ATF4 and GADD34 **c** and its intensity measurements **d** in age-matched normal and glaucoma donor eyes shows significantly increased ATF4 and GADD34 in glaucomatous TM tissues. Arrow shows TM. SC Schlemm's canal. Scale bar is 100 μm ($n = 9$ normal and 11 glaucoma eyes, data are presented as mean ± SEM, two-tailed unpaired $t$-test). *Indicates fold in section (artifact). **e** Periocular Dex injections elevate IOP significantly in C57BL/6J mice. 3 months old C57BL/6J mice received vehicle or Dex (200 μg/eye) via periocular conjunctival fornix injections bilaterally every week up to 5 weeks and night IOPs were monitored weekly using rebound tonometry ($n = 8$ in each group; data are presented as mean ± SEM, two-way ANOVA, ***$P < 0.001$). **f** and **g** Western blot for ER stress markers **f** and its densitometric analyses **g** of anterior segment tissues from vehicle and Dex-treated mice demonstrated that induction of chronic ER stress markers is associated with Dex-induced ocular hypertension ($n = 3$ in each group, data are presented as mean ± SEM, two-tailed unpaired $t$-test).

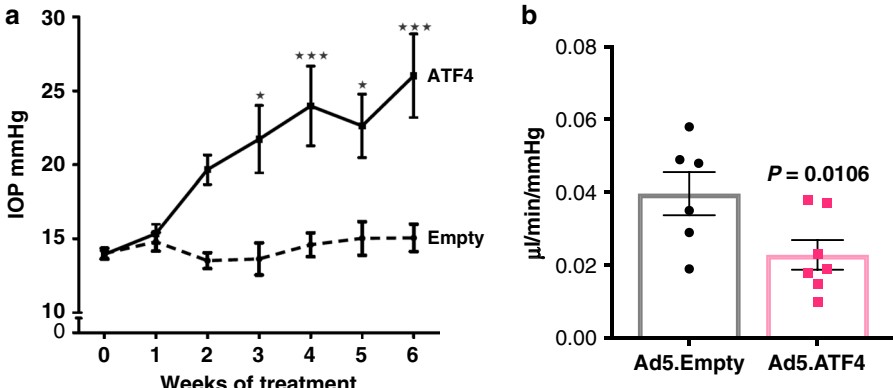

**Fig. 2 ATF4 induces ocular hypertension and reduces aqueous humor outflow facility in mice. a** 3 months old C57BL/6J mice received bilateral intravitreal injections of Ad5.Empty or Ad5.ATF4 ($2 \times 10^7$ pfu/eye). IOPs were monitored using rebound tonometry on conscious mice. ATF4-injected mice showed significant IOP elevation starting from 3 weeks injections compared to Empty-injected mice ($n = 10$ each group, data are presented as mean ± SEM, two-way ANOVA, Bonferroni's multiple comparisons test *$P < 0.05$, ***$P < 0.001$). **b** Conventional outflow facility was significantly reduced in Ad5.ATF4-injected mice (0.022 μL/min/mmHg; $n = 6$) compared to Ad5.Empty injected mice (0.042 μL/min/mmHg; $n = 6$) (data are presented as mean ± SEM, two-tailed unpaired $t$-test).

6 weeks (Fig. 2a). A significant and sustained IOP elevation was observed in ATF4 injected mice compared to Ad5.Empty injected mice starting at 3 weeks. We also observed that adenoviral expression of GFP did not elevate IOP significantly (Supplementary Fig. 5). We next examined whether ATF4-mediated IOP elevation is due to reduced outflow facility by measuring outflow facility using a constant flow infusion method[54,55]. A significant

decrease in outflow facility (0.022 μl/min/mmHg in Ad5.ATF4 vs. 0.042 μl/min/mmHg in Ad5.Empty; $p < 0.0106$) was observed in Ad5.ATF4 injected mice compared to Ad5.Empty injected mice 5 weeks after injection (Fig. 2b). Immunostaining of ATF4 in the anterior segment tissues revealed prominent localization of ATF4 in TM tissues of mice injected with Ad5.ATF4 while mice injected with Ad5.Empty showed little expression of ATF4

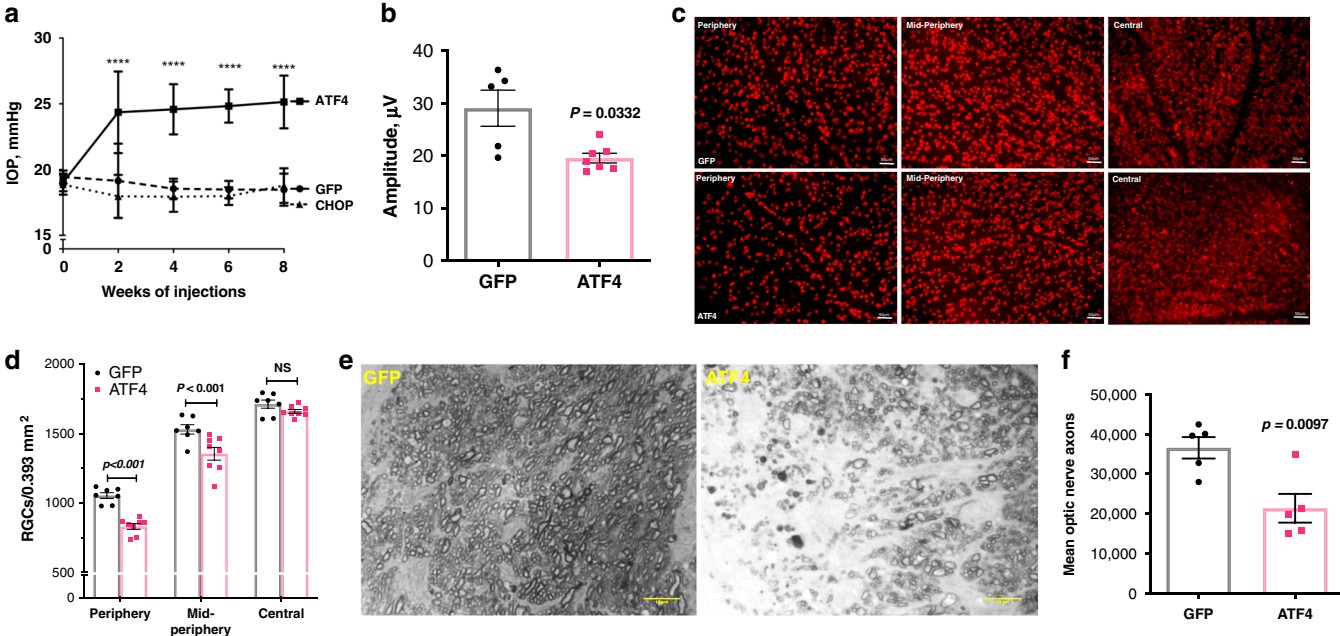

**Fig. 3 Lentiviral expression of ATF4 but not CHOP induces glaucoma in mice. a** Night-time IOP measurements in mice injected intravitreally with lentiviral particles expressing GFP (control), ATF4, and CHOP ($2 \times 10^6$ TU/eye). A significant IOP elevation was observed in ATF4-injected mice compared to GFP or CHOP-injected mice ($n = 7$ in GFP group, $n = 8$ in ATF4 and CHOP groups; data are presented as mean ± SEM, two-way ANOVA, Tukey's multiple comparison test, ****$P < 0.0001$). **b** RGC functional loss in ATF4-injected mice. A decrease in pattern ERG amplitudes was observed in 8 weeks ATF4-injected mice ($n = 7$) compared to GFP-injected mice ($n = 5$; data are presented as mean ± SEM, two-tailed unpaired $t$-test. **c** Loss of RGCs in 10 weeks ATF4-injected mice. Representative images showing peripheral, mid-peripheral, and central regions of whole mount retinas stained with RBPMS (RGC marker) in GFP ($n = 7$) and ATF4 ($n = 8$) injected groups. **d** RGC counts per 0.393 mm$^2$ of total retina in peripheral, mid-peripheral, and central regions of ATF4 and GFP-injected mice demonstrated significant loss of RGCs in periphery and mid-periphery of ATF4-injected retinas. Scale bar is 50 μm. Data are presented as mean ± SEM, one-way ANOVA; $n = 7$ for GFP and $n = 8$ for ATF4. **e** Representative images of optic nerve sections stained with PPD showing loss of axons in 10 weeks ATF4-injected mice. Scale bar is 10 μm ($n = 5$ in GFP and $n = 5$ in ATF4 injected mice). **f** PPD stained optic nerve axons were counted and total number of axons per optic nerve were shown graphically in GFP ($n = 5$) vs. ATF4 ($n = 5$) injected mice groups (data are presented as mean ± SEM, two-tailed unpaired $t$-test).

(Supplementary Fig. 6). We also observed that adenoviral injections of Empty and ATF4 caused mild to moderate ocular inflammation, respectively (Supplementary Fig. 7). Although IOP elevation was sustained, examination of other glaucoma features including functional analysis of RGCs was not possible due to ocular inflammation.

**Expression of ATF4 leads to glaucomatous neurodegeneration.** Previous studies have shown that lentiviral particles cause minimum ocular inflammation and have selective tropism for TM cells in vivo[56–58]. We therefore generated lentiviral particles expressing ATF4 and CHOP and further examined whether ATF4-induced IOP elevation leads to glaucomatous neurodegeneration in mice. Two-months old C57BL/6J mice were injected intravitreally with lentiviral particles ($2 \times 10^6$ TU/eye) expressing GFP, ATF4, or CHOP. IOP measurements (night and day) were performed weekly (Fig. 3a and Supplementary Fig. 8). Similar to Ad5.ATF4 effects on IOP, we observed that lentiviral particles expressing ATF4 led to significant and sustained IOP elevation. Interestingly, we observed that lentiviral expression of CHOP did not elevate IOP significantly compared to mice injected with GFP. Similar results were obtained in Ad5.CHOP injected mice (data not shown) indicating that expression of CHOP alone is not sufficient to elevate IOP in mice. Immunostaining of ATF4 and CHOP demonstrates expression of ATF4 and CHOP in TM tissues of lentiviral injected mice (Supplementary Fig. 9). H & E staining (Supplementary Fig. 10) reveals no ocular toxicity and open iridocorneal angles in GFP, ATF4, and CHOP injected mice.

We next examined whether ATF4 leads to functional and structural loss of RGCs, as is observed in glaucoma patients. We examined functional deficits in RGCs using pattern electroretinography (PERG)[59]. After 8 weeks of injections, ATF4-injected mice demonstrate about 32% reduction in PERG amplitude compared to mice injected with GFP lentiviral particles (Fig. 3b). We next performed whole mount staining of retinas using the RGC-specific marker, RBPMS[60]. As shown in representative images, ATF4 lentiviral-injected retinas demonstrate loss of RBPMS-positive RGCs (Fig. 3c and Supplementary Fig. 11). Quantitation of RBPMS-positive cells revealed significant loss of RGCs in the periphery (21%) and mid-periphery (12%), but not in the central retinas of 10-weeks ATF4-injected mice compared to the retinas of GFP-injected mice (Fig. 3c, d).

We next examined whether ATF4 leads to optic nerve degeneration by staining optic nerve cross-sections with paraphenylenediamine (PPD) and quantified optic nerve axons. As shown in representative images, optic nerve degeneration is evident in ATF4-injected mice (Fig. 3e). Quantitation of optic nerve axons demonstrated a significant loss of axons (42%) in 10-weeks ATF4-injected mice compared to GFP-injected mice (Fig. 3f). These data indicate that ATF4 leads to glaucomatous neurodegeneration in mice.

**CHOP is required for ATF4-induced IOP elevation.** Both ATF4 and CHOP are induced in glaucomatous TM tissues. However, expression of ATF4 alone but not CHOP elevates IOP significantly in mice. Since ATF4 is upstream of CHOP in the ATF4–CHOP pathway, we hypothesized that ATF4 induces

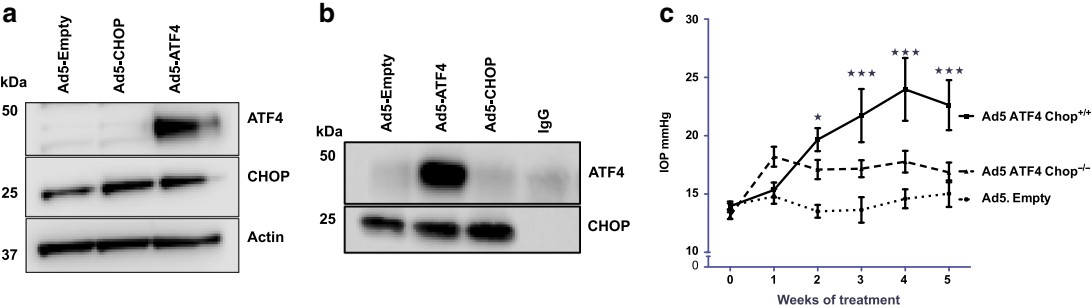

**Fig. 4 CHOP is required for ATF4-induced IOP elevation. a** GTM3 cells were transduced with Ad5.Empty, ATF4, and CHOP for 24 h. Cellular lysates were subjected to Western blot analysis of ATF4 and CHOP. Increased CHOP expression levels were observed in ATF4-transduced TM cells whereas CHOP transduction did not alter ATF4 levels in TM cells ($n = 3$ replicates). **b** GTM3 cells transduced with Ad5.Empty, ATF4, and CHOP for 24 h. Cellular lysates were immunoprecipitated using CHOP antibody and immunoblotted for ATF4. IgG antibody was used as a negative control ($n = 3$ replicates). Experiments were replicated three times independently and similar results were observed in NTM-5 cell line. **c** 3-month-old WT (CHOP$^{+/+}$) and CHOP knockout (CHOP$^{-/-}$) mice were intravitreally injected with Ad5.Empty or ATF4. A significant IOP elevation was observed in CHOP$^{+/+}$ mice injected with Ad5.ATF4 ($n = 8$) compared to CHOP$^{+/+}$ mice injected with Ad5.Empty ($n = 6$). However, ATF4 did not significantly elevate IOP in CHOP$^{-/-}$ mice ($n = 8$) compared to Ad5. Empty injected WT mice (data are presented as mean ± SEM, two-way ANOVA with Bonferroni's multiple comparison test, *$P < 0.05$, ***$P < 0.001$).

CHOP expression, and physical interaction of ATF4 with CHOP is required for IOP elevation. We first examined whether expression of ATF4 induces CHOP in GTM3 cells. GTM3 cells were transduced with Ad5.Empty, Ad5.ATF4, or Ad5.CHOP at 100 MOI for 24 h. Western blot analysis (Fig. 4a) revealed that expression of ATF4 increases CHOP expression levels, while expression of CHOP did not alter ATF4 levels in GTM3 cells. We next explored whether ATF4 interacts with CHOP in GTM3 cells by co-immunoprecipitation studies (Fig. 4b). GTM3 cells were transduced with Ad5.Empty, Ad5.CHOP, or Ad5.ATF4 for 24 h. Total cell lysates were immunoprecipitated using anti-CHOP antibody and blotted for ATF4. Lysates incubated with IgG antibody, which showed no bands for CHOP served as a negative control. A strong interaction between CHOP and ATF4 is observed only in ATF4 transduced TM cells compared to Empty and CHOP-transduced cells. To further clarify whether the ATF4–CHOP interaction is required for IOP elevation, we performed adenoviral intravitreal injections of ATF4 in WT and *Chop* knockout mice. Adenoviral expression of ATF4 in *Chop*$^{+/+}$ mice significantly elevated IOP at 2, 3, 4, and 5 weeks post-injection compared to Ad5.Empty injected *Chop*$^{+/+}$ mice (Fig. 4c). However, expression of ATF4 did not elevate IOP significantly in *Chop*$^{-/-}$ mice. These data demonstrate that CHOP is essential for ATF4-induced IOP elevation.

**ATF4 increases protein synthesis and ER client protein load.** We further explored how ATF4 leads to TM damage and IOP elevation. We first examined ER stress markers in GTM3 cells transduced with Ad5.Empty (E), ATF4, and CHOP. Western blot analysis demonstrated that expression of ATF4 but not CHOP induced spliced XBP-1 indicating that expression of ATF4 leads to induction of ER stress (Fig. 5a). Interestingly, ATF4 also induced GADD34 and decreased phosphorylation of eIF2α. Increased GADD34 and decreased phosphorylated eIF2α suggest that ATF4 may lead to increased protein synthesis. A previous study has shown that ATF4-induced protein synthesis is associated with cell death[51]. We have shown that Dex increases general protein synthesis and induces secretory protein overload in TM cells[37]. We therefore hypothesized that induction of ATF4 leads to increased protein synthesis and ER client protein load, which may be associated with TM cell dysfunction and death. We examined de novo protein synthesis in TM cells using the SUnSET assay, a nonradioactive method of detecting de novo protein synthesis[37]. GTM3 cells were transfected with plasmids expressing Empty, ATF4, or CHOP for 48 h and incubated with

puromycin for 30 min. Total lysates were subjected to Western blot analysis using anti-puromycin antibody. ATF4 led to increased protein synthesis, which was blocked by pre-treatment with cyclohexamide (CHX) (10 μg/ml) (Fig. 5b). Interestingly, expression of CHOP did not alter protein synthesis compared to control cells. Since TM cells are highly secretory, we explored whether ATF4-induced protein synthesis further leads to increased ER client protein load. GTM3 cells were transfected with plasmids expressing Empty, ATF4, or CHOP for 48 h and incubated with puromycin for 30 min. Total ER was isolated and equal protein from ER lysates were subjected to Western blot analysis using anti-puromycin antibody (Supplementary Fig. 12). ATF4 increased de novo protein load in the ER and also increased ER chaperones suggesting that expression of ATF4 leads to increased ER protein client load and induces ER stress in TM cells. MYOC is a secreted protein and it is highly expressed by TM cells[61–63]. We examined whether expression of ATF4 leads to increased accumulation of WT and mutant MYOC in TM cells. GTM3 cells stably expressing DsRed-tagged WT or mutant (Y437H) MYOC were transfected with plasmids expressing ATF4 for 48 h and immunostained with an ER marker, calreticulin (Fig. 5c, d), or ER stress marker, KDEL (Supplementary Fig. 13A). We observed that in control cells, WT MYOC is mostly secreted with the presence of little intracellular MYOC. Interestingly, expression of ATF4 increased WT MYOC in the ER of GTM3 cells as evident from increased co-localization with the ER marker, calreticulin (Fig. 5c). This increased intracellular accumulation of WT MYOC resulted in induction of ER stress as evident from increased KDEL staining (Supplementary Fig. 13A). KDEL antibody recognizes GRP78 and GRP94 in TM cells[37,64]. Furthermore, expression of ATF4 aggravated mutant MYOC accumulation and its co-localization with calreticulin in GTM3 cells stably expressing DsRed-tagged mutant MYOC (Fig. 5d). Western blot analysis of triton insoluble form of MYOC revealed that there is little or no insoluble WT MYOC in GTM3 cells expressing WT MYOC. However, expression of ATF4 increased triton insoluble WT MYOC similar to that of mutant MYOC levels (Supplementary Fig. 13B). These data further indicate that expression of ATF4 leads to increased ER client protein load in TM cells.

We next examined whether expression of ATF4 leads to ER stress in the TM in vitro and in vivo. Primary human TM cells ($n = 3$ cell strains) were transduced with Ad5.Empty, Ad5.ATF4, Ad5.ATF4ΔRK (dominant negative inhibitor of ATF4), or Ad5.CHOP at 50 MOI for 3 days. Cell lysates were subjected to

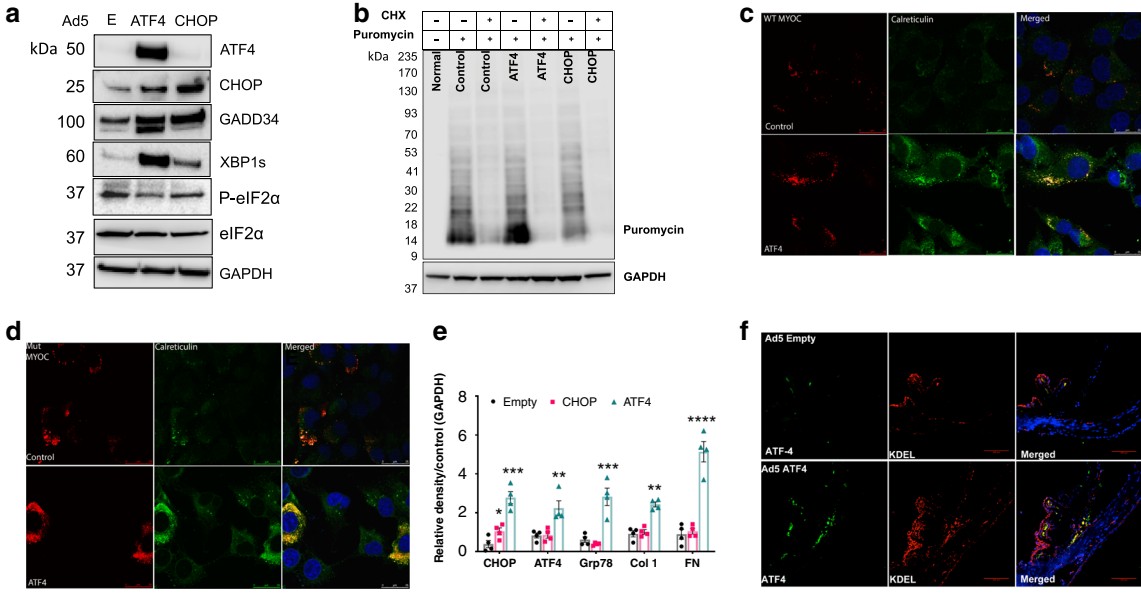

**Fig. 5 ATF4 increases protein synthesis and ER client protein load in TM cells and tissues. a** GTM3 cells were transduced with Ad5.Empty, ATF4, and CHOP for 36 h. and cellular lysates were examined for markers of chronic ER stress by Western blot analysis. Expression of ATF4 increased XBP-1 (s) and GADD34 while ATF4 reduced p-eIF2α ($n = 3$ replicates). **b** GTM3 cells transfected with ATF4 or CHOP were incubated with cycloheximide (CHX) (10 μg/ml) for 16 h. Puromycin (10 μg/ml) was added to cells for 30 min before harvesting cell lysates. Total cellular lysates were subjected to Western blot analysis using anti-puromycin and GAPDH antibodies. Increased puromycin incorporation observed in the total ER fractions of ATF4 transfected cells ($n = 3$ replicates), which was blocked by CHX signifying a higher rate of de novo protein synthesis. **c** and **d** GTM3 cells stably expressing DsRed-tagged WT **c** or mutant **d** MYOC were transfected with plasmids expressing Empty and ATF4 for 48 h. Cells were fixed and stained with calreticulin (ER marker). ATF4 increased intracellular WT and mutant myocilin accumulation in the ER as evident from a strong colocalization of myocilin with calreticulin in ATF4-transfected cells ($n = 3$ replicates). Scale bar is 25 μm. **e** 3-month-old C57BL/6J mice were intravitreally injected with Ad5.Empty, Ad5.ATF4, and Ad5.CHOP ($2 \times 10^7$ pfu/eye). After 3 weeks of injections, anterior segment lysates were subjected to Western blot (Supplementary Fig. 15) and densitometric analyses **e** of ECM (fibronectin, collagen-1) and ER stress (Grp78, ATF4, and CHOP) markers ($n = 4$ mice; data are presented as mean ± SEM, one-way ANOVA, *$P < 0.05$, **$P < 0.01$, ***$P < 0.001$, ****$P < 0.0001$). **f** Increased ATF4 and KDEL staining in TM of Ad5.ATF4-injected mice compared to Ad5. Empty-injected mice ($n = 3$ mice). Scale bar is 100 μm.

Western blot analysis of ECM and ER stress markers (Supplementary Fig. 14). Adenoviral expression of ATF4 increased fibronectin, GRP78, GRP94, and CHOP compared to Ad5. Empty-treated TM cells. Transduction of TM cells with ATF4ΔRK prevented induction of fibronectin and ER stress markers. Interestingly, expression of CHOP did not alter ECM and ER stress markers. Western blot analysis further confirmed that adenoviral expression of ATF4, ATF4ΔRK, and CHOP increases protein levels of ATF4 and CHOP, as expected. We also examined whether ATF4-induced ocular hypertension is associated with induction of ER stress in vivo as observed in human glaucomatous TM tissues. Three-month-old animals were administered intravitreal injections of Ad5.Empty, Ad5.ATF4, or Ad5.CHOP at $2 \times 10^7$ pfu/eye. Western blot (Supplementary Fig. 15) and densitometric analysis (Fig. 5e) revealed that expression of ATF4, but not CHOP, significantly increases ECM markers including fibronectin and type 1 collagen (Col1). Increased CHOP and GRP78 accompanied by increased ATF4 levels were observed in the Ad5.ATF4-injected group compared to controls. Interestingly, no changes in ECM and ER stress markers were observed in the Ad5.CHOP-injected group compared to the Ad5.Empty control group. We further confirmed increased ATF4 and KDEL in mouse TM using immunostaining (Fig. 5f). ATF4 co-localized with the ER stress marker in TM tissues only in ATF4-injected mice.

**ATF4 leads to TM cell death in vitro and in vivo.** Since the ATF4–CHOP pathway is associated with cell death, we examined whether ATF4 leads to TM cell death in GTM3 cells. GTM3 cells

were transduced with Ad5.Empty, Ad5.CHOP, or Ad5.ATF4 for 36 h. Cellular lysates were utilized for Western blot analysis of apoptotic markers (Fig. 6a) and fixed cells were analyzed by TUNEL assay (Fig. 6b). Cleaved Caspase 3 and cleaved PARP (the classical markers of apoptosis) were observed only in Ad5.ATF4-transduced GTM3 cells, but not in Ad5.CHOP-transduced GTM3 cells (Fig. 6a). Surprisingly, ATF4 did not alter death receptor 5 (DR5), which is previously shown to be involved in ER stress-induced cell death[65]. We observed TUNEL-positive cells only in Ad5.ATF4-transduced GTM3 cells (Fig. 6b) and in NTM5 cells (Supplementary Fig. 16). Increased numbers of TUNEL-positive cells were observed in TM tissues of Ad5.ATF4-injected mice but not in CHOP-injected mice (Fig. 6c). Together, these data demonstrate that expression of ATF4 but not CHOP induces apoptosis of TM cells in vitro and in vivo. Interestingly, ATF4 did not alter the expression of other pro-apoptotic markers including BAX, BAD, and DR5 suggesting that ATF4-induced TM cell death is independent of these pro-death proteins.

We next examined whether ATF4-induced apoptosis of TM cells is dependent on CHOP. GTM3 cells were transduced with Ad5.Empty, Ad5.ATF4 alone, Ad5.ATF4 plus Ad5.CRISPR targeting CHOP (CR-CHOP) or Ad5.CHOP alone. As expected, GTM3 cells transduced with ATF4 alone significantly increased cleaved PARP while knock down of CHOP significantly rescued induction of cleaved PARP compared to GTM3 cells transduced with ATF4 alone (Fig. 6d, e). These data indicate that ATF4-induced TM cell death is dependent on CHOP. We next examined whether ATF4-induced protein synthesis is involved in TM cell death. Treatment with a mild dose of CHX (2.5 μg/ml) significantly reduced ATF4-induced cleaved PARP levels without

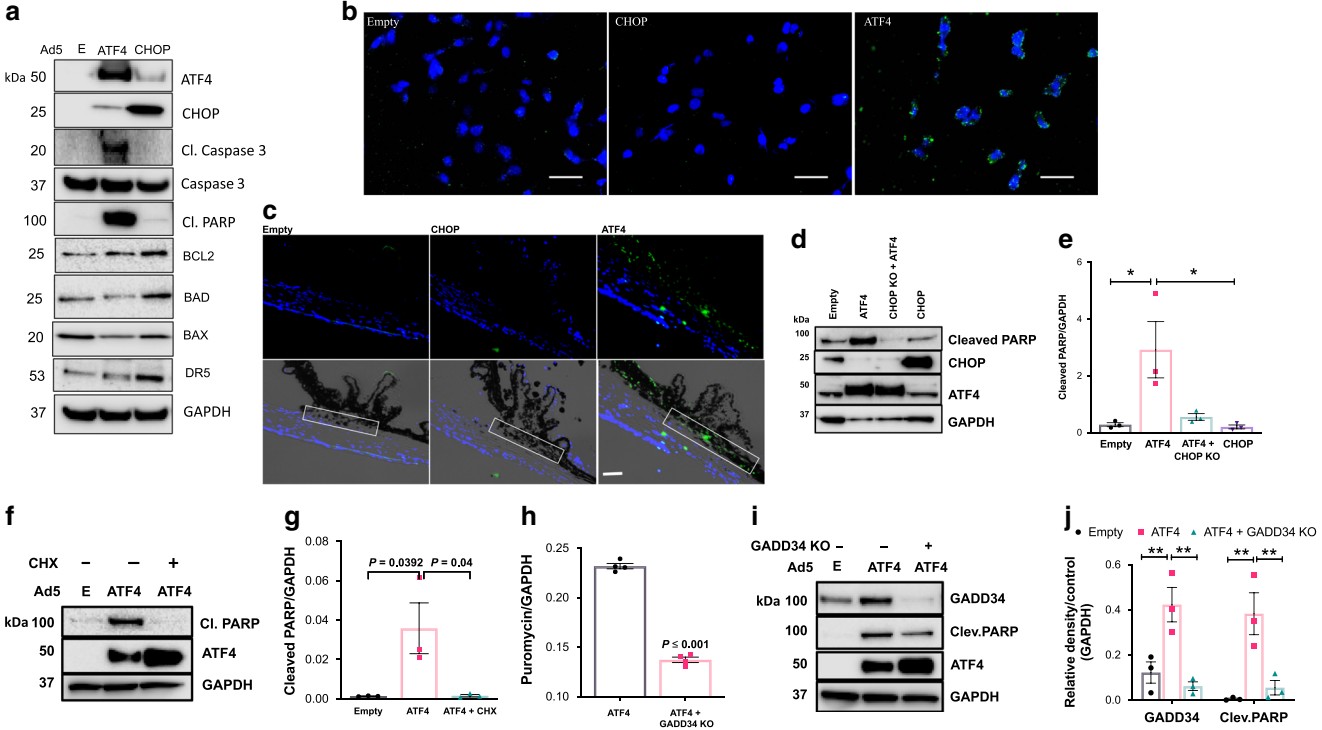

**Fig. 6 ATF4 leads to TM cell death in a CHOP and GADD34-dependent manner.** GTM3 cells were transduced with Ad5.Empty, ATF4, or CHOP for 36 h.
**a** Cellular lysates were subjected to Western blot analysis for apoptotic markers ($n = 3$ independent experiments). **b** Fixed GTM3 cells transduced with
Ad5.Empty, ATF4, or CHOP were analyzed by TUNEL assay. Increased number of TUNEL-positive cells (green) were observed in Ad5.ATF4-transduced
TM cells ($n = 3$ independent experiments). Scale bar is 50 μm. **c** Increased number of TUNEL-positive cells (green) were observed in the TM region of
2 weeks Ad5.ATF4-injected mice compared to Ad5.Empty or Ad5.CHOP ($n = 3$ in each groups). Bright field images (bottom panel) merged with DAPI
shows TM orientation and the TM area is represented in a white box (scale bar = 50 μm). **d** and **e** GTM3 cells were transduced with Ad5.Empty, ATF4,
CHOP, or ATF4 + CHOP KO (CRISPR-Cas9 vector targeting CHOP) for 36 h. Western blot **d** and densitometric analyses **e** demonstrated that depletion of
CHOP significantly reduced ATF4-induced cleaved PARP suggesting that CHOP is required for ATF4-mediated cell death. $n = 3$ independent experiments,
data are presented as mean ± SEM, *$p < 0.05$, one-way-ANOVA. **f** and **g** Western blot **f** and densitometric analyses **g** of cleaved PARP in GTM3 cells
transduced with ATF4 and treated with mild dose of CHX (2.5 μg/ml) demonstrated that reduction of protein synthesis prevents ATF4-induced TM cell
death. $n = 3$ independent experiments, data are presented as mean ± SEM, one-way-ANOVA. **h** GTM3 cells expressing ATF4 were transfected with
plasmid expressing CRISPR-Cas9 targeting GADD34. Puromycin (10 μg/ml) was added to cells for 30 min before harvesting cell lysates. Total cellular
lysates were subjected to Western blot analysis using anti-puromycin and anti-GAPDH antibodies (shown in Supplementary Fig. 17). Densitometric
analysis demonstrated that GADD34 knock down significantly reduces ATF4-induced protein synthesis. $n = 4$ independent experiments, data are
presented as mean ± SEM, two-tailed unpaired $t$-test. **i** and **j** Western blot **i** and densitometric analyses **j** of cleaved PARP and GADD34 of GTM3 cells
expressing ATF4 and plasmid expressing CRISPR-Cas9 targeting GADD34 demonstrated that knockdown of GADD34 significantly reduced ATF4-
induced cleaved PARP and GADD34 protein levels. $n = 3$ independent experiments, data are presented as mean ± SEM, **$p < 0.01$, two-way-ANOVA.

altering ATF4 levels (Fig. 6f, g). Of note, treatment with CHX
prominently reduced ATF4-induced protein synthesis (Fig. 5b).
Since ATF4-induced GADD34 is known to be involved in
promoting protein synthesis, we further examined whether
depletion of GADD34 rescues ATF4-induced protein synthesis
(Fig. 6h and Supplementary Fig. 17) and TM cell death (Fig. 6i, j).
Interestingly, depletion of GADD34 prominently reduced ATF4-
induced protein synthesis in GTM3 cells (Fig. 6h). Furthermore,
depletion of GADD34 significantly reduced ATF4-induced
cleaved PARP in GTM cells (Fig. 6i, j). These studies clearly
indicate that ATF4-induced TM cell death is dependent on
protein synthesis and expression of CHOP and GADD34.

Several studies have linked the role of ATF4 in oxidative
stress[66–68]. We therefore examined whether ATF4 leads to
oxidative stress in TM cells using the DCFDA assay. A significant
increase in cellular reactive oxygen species (ROS) was observed
upon overexpression of ATF4 in GTM3 cells (Supplementary
Fig. 18A). Treatment with a chemical chaperone, sodium 4-
phenylbutyrate (PBA) or an antioxidant, butylated hydroxyani-
sole (*BHA*) significantly reduced cellular ROS levels induced by
ATF4. Since ATF4-induced cell death has been shown to be

associated with oxidative stress[51,66], we examined whether ATF4-
induced oxidative stress is a causative factor in ATF4-induced
apoptosis in GTM cells. GTM3 cells transduced with Ad5.ATF4
with or without PBA or BHA (Supplementary Fig. 18B)
demonstrated that depletion of oxidative stress by PBA or BHA
does not rescue TM cells from ATF4-induced apoptosis.

Increased TM and SC stiffness is associated with increased
outflow resistance and IOP elevation in glaucoma[69–72]. Cell–ECM
interactions are predominant drivers of tissue stiffness. Since
chronic ER stress markers were only upregulated in TM region
(Fig. 1c), we further explored whether expression of ATF4 leads to
altered expression of actin-regulating proteins in human TM cells
and mouse TM tissues. Adenoviral expression of ATF4 increased
F-actin filaments and α-smooth muscle actin (α-SMA) in mouse
TM tissues (Supplementary Fig. 19). Moreover, we observed that
ATF4-induced expression of cleaved Rho-associated coiled coil-
containing protein kinase (ROCK)-1 and phosphorylation of
myosin light chain (MLC)2 in NTM5 cells (Supplementary
Fig. 20A–C). Furthermore, expression of ATF4 significantly
increased phosphorylation of MLC2 in mouse anterior segment
lysates (Supplementary Fig. 20D, E). These studies suggest that

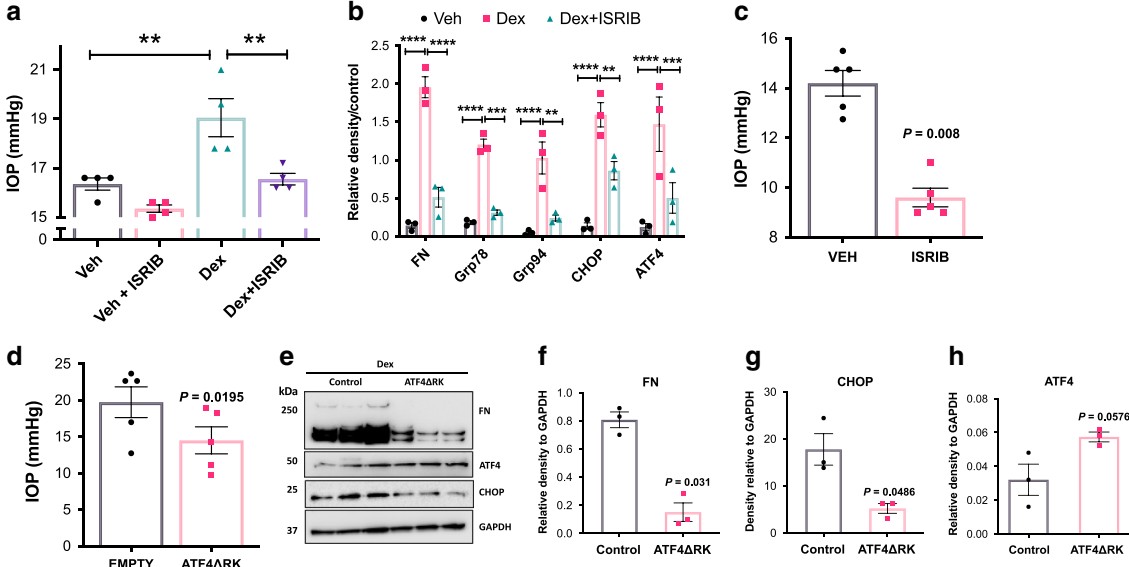

**Fig. 7 Pharmacological inhibition of ATF4 rescues mouse models of glaucoma. a** C57 mice were injected with vehicle ($n = 4$) and Dex ($n = 4$) weekly via periocular conjunctival fornix injections. After 2 weeks, left eyes received 5 μl of 2 mM ISRIB topical eye drops while the contralateral right eyes received vehicle eye drops (DMSO) twice daily. One-week ISRIB treatment significantly lowered IOPs in Dex-injected left eyes compared to the contralateral right eyes ($n = 4$ biologically independent samples; data are presented as mean ± SEM, one-way ANOVA, Tukey's multiple comparison test, $**P < 0.01$). **b** Western blot (Supplementary Fig. 21) and densitometric analysis **b** of mouse anterior segment lysates injected with vehicle or Dex for 2 weeks and treated with 5 μl of 2 mM ISRIB topical eye drops or DMSO control. ISRIB treatment prevented Dex-induced FN and ER stress markers ($n = 3$ biologically independent samples; data are presented as mean ± SEM, one-way ANOVA, $**P < 0.01$, $***P < 0.001$, $****P < 0.0001$). **c** The ocular hypertensive *Tg. MYOC^{Y437H}* mice received ISRIB eye drops in left eyes whereas the contralateral right eyes received vehicle (DMSO) eye drops twice daily. IOPs were recorded after one-week treatment ($n = 5$ biologically independent samples; data are presented as mean ± SEM, two-tailed paired *t*-test). **d** C57 mice were treated with Dex for 3 weeks via periocular route. IOPs were measured after 3 weeks. After ocular hypertension was confirmed, Ad5.ATF4ΔRK ($2 \times 10^7$ pfu/eye) was injected intravitreally in the left eyes while the contralateral right eyes were injected with Ad5.Empty. Both eyes were treated with Dex for another week and IOPs were measured every week. Ad5.ATF4ΔRK reduced elevated IOP compared to contralateral eyes injected with Ad5. empty ($n = 4$ biologically independent samples; data are presented as mean ± SEM, two-tailed paired *t*-test). **e–h** Western blot **e** and its densitometric analyses **f–h** of mouse anterior segment lysates show that ATF4ΔRK prevented Dex-induced fibronectin and CHOP. $n = 3$ biologically independent samples; data are presented as mean ± SEM, two-tailed unpaired *t*-test.

ATF4 modulates contractile functions of TM, which may contribute to increasing outflow resistance.

**Inhibition of ATF4–CHOP pathway rescues mouse models of glaucoma.** Recent studies have identified a small molecule N, N′-trans-(cyclohexane-1,4-diyl)-bis-(2-(4-chlorophenoxy) acetamide (ISRIB), as a selective and potent inhibitor of the ATF4/CHOP pathway[73,74]. ISRIB acts downstream of elF2α and upstream of the ATF4–CHOP pathway[75]. We examined whether inhibition of the ATF4–CHOP pathway with ISRIB reduces elevated IOP in ocular hypertensive Dex-treated or *Tg-MYOC^{Y437H}* mice. C57 mice (3 months old) were treated with weekly periocular injections of vehicle or Dex, and IOPs were monitored (Fig. 7a). Within 2 weeks of treatment, Dex significantly elevated IOP compared to vehicle-treated mice. At this stage, topical ocular ISRIB eye drops (5 μl of 2 mM stock) were given to left eyes, while the contralateral right eyes received vehicle eye drops (DMSO dissolved in PBS). IOP measurements after 1-week treatment revealed that ISRIB significantly reduces IOP in Dex-treated mice (Fig. 7a). Anterior segment tissue lysates collected from mice treated with Veh, Dex, and Dex + ISRIB were subjected to Western blot analysis of various markers of the ECM and the ER stress pathway (Supplementary Fig. 21 and Fig. 7b). Densitometric analysis confirmed that Dex significantly increases ECM and ER stress markers. ISRIB significantly reduced Dex-induced ATF4 and CHOP as well as the level of ECM and ER stress markers (Fig. 7b).

We next examined whether reduction of the ATF4–CHOP pathway reduces elevated IOP in the mouse model of MYOC-associated POAG using transgenic mice expressing mutant human MYOC (*Tg-MYOC^{Y437H}* mice). We have previously shown that *Tg-MYOC^{Y437H}* mice develop ocular hypertension starting at 3 months of age and that mutant MYOC-induced ocular hypertension is associated with chronic ER stress[38,76]. To examine whether ISRIB reduces elevated IOP in *Tg-MYOC^{Y437H}* mice, the ocular hypertensive 4-month-old *Tg-MYOC^{Y437H}* mice were given topical ocular ISRIB eye drops (2 mM) in the left eye, while the contralateral right eye received vehicle eye drops (Fig. 7c). IOP measurement after 1-week revealed that ISRIB significantly reduces elevated IOP in *Tg-MYOC^{Y437H}* mice.

Previous studies have shown that the dominant-negative inhibitor of ATF4 (ATF4ΔRK) inhibits transcriptional activity of endogenous *ATF4*[66]. We examined whether expression of ATF4ΔRK rescues Dex-induced ocular hypertension (Fig. 7d). C57BL/6J mice (3 months old) were treated with weekly periocular injections of Dex, and IOPs were monitored. After 3 weeks of Dex treatment, the left eye of Dex-treated ocular hypertensive mice were intravitreally injected with Ad5.ATF4ΔRK while the contralateral right eyes received Ad5.Empty. IOP measurements after 1 week of treatment revealed that ATF4ΔRK significantly reduces elevated IOP in Dex-treated mice (Fig. 7d). Western blot (Fig. 7e) and densitometric analyses (Fig. 7f–h) revealed that Ad5.ATF4ΔRK significantly reduces Dex-induced CHOP and fibronectin levels.

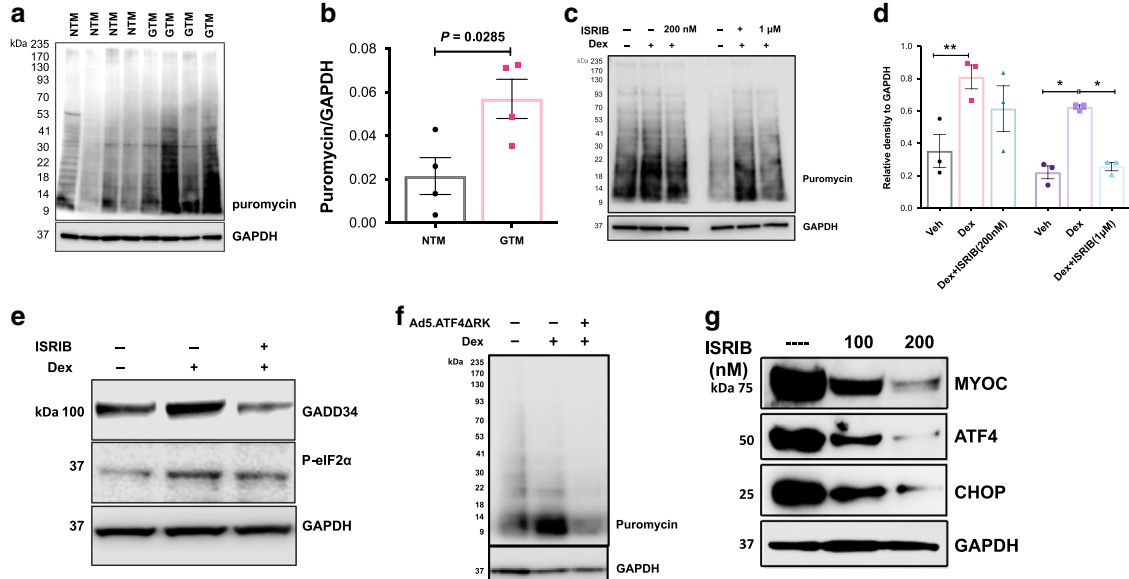

**Fig. 8 Increased protein synthesis is associated glaucomatous primary human TM cells. a** and **b** Protein synthesis was examined in human primary TM cells from normal (*n* = 4 biologically independent cells) and POAG donor eyes (*n* = 4 biologically independent cells) using SUnSET assay. Western blot **a** and its densitometric analysis **b** demonstrated significantly increased protein synthesis in glaucomatous TM cells. *n* = 4 biologically independent samples, data are presented as mean ± SEM, two-tailed unpaired *t*-test. **c** and **d** GTM3 cells were treated with vehicle and Dex along with DMSO (control) or ISRIB for 2 h. Protein synthesis was examined using the SUnSET assay. Western blot **c** and its densitometric analysis **d** demonstrated ISRIB significantly reduced Dex-induced protein synthesis. *n* = 3 independent experiments, data are presented as mean ± SEM, \*\**p* < 0.01; \**p* < 0.05; 1-way ANOVA. **e** Western blot analysis demonstrated that ISRIB reduced Dex-induced GADD34 levels while ISRIB effects on p-eIF2α were minimal. *n* = 3 independent experiments. **f** Western blot analysis of total ER fractions with puromycin antibody isolated from GTM3 cells treated with vehicle, Dex, and Dex plus ATF4ΔRK for 36 h. Cells were incubated with puromycin antibody for 30 min before harvesting lysates (*n* = 3 independent experiments). Genetic inhibition of ATF4 prevented Dex-induced ER client protein load in GTM3 cells. **g** Western blot analysis of GTM3 cells expressing mutant myocilin treated with ISRIB showing that ISRIB reduces intracellular accumulation of mutant myocilin along with ATF4 and CHOP protein levels. *n* = 3 independent experiments.

**Increased protein synthesis in glaucomatous human TM cells**. Since ATF4–CHOP–GADD34 are induced in glaucomatous TM cells, we further explored whether increased protein synthesis is associated with human primary TM cells obtained from POAG donors. Age-matched human primary TM cells from normal and POAG donor eyes (*n* = 4 cell strains each) were subjected to the SUnSET assay (Fig. 8a, b). Glaucomatous TM cells clearly demonstrated significantly increased protein synthesis compared to normal TM cells. We also explored whether ISRIB reduces Dex-induced protein synthesis in GTM3 cells. GTM3 cells were treated with vehicle or Dex with or without ISRIB. GTM3 cells were incubated with puromycin for 30 min before harvesting cell lysates. Treatment of GTM3 cells with ISRIB significantly reduced Dex-induced protein synthesis (Fig. 8c, d). Western blot analysis further revealed that ISRIB inhibits GADD34 and its effect on Dex-induced phosphorylation of eIF2α is minimal (Fig. 8e). We next examined whether genetic depletion of ATF4 reduces Dex-induced ER client protein load. GTM3 cells were treated with vehicle or Dex with or without Ad5.ATF4ΔRK. GTM3 cells were incubated with puromycin for 30 min before harvesting cell lysates. ER isolates were subjected to Western blot analysis using an anti-puromycin antibody (Fig. 8f). Dex increased ER client protein load, which was reduced by Ad5.ATF4ΔRK indicating that ATF4 depletion reduces Dex-induced protein synthesis and ER client protein load in TM cells. Treatment of GTM3 cells stably expressing mutant MYOC with ISRIB reduced both ATF4 and CHOP, as well as reduced intracellular mutant MYOC accumulation (Fig. 8g) indicating that ISRIB reduces ER client protein load. These studies support that increased protein synthesis is associated with induction of chronic ER stress-induced ATF4–CHOP–GADD34 pathway and targeting this pathway rescues mouse model of glaucoma.

## Discussion

Chronic ER stress is associated with TM dysfunction and IOP elevation in mouse and human glaucoma. However, it is not understood how chronic ER stress leads to TM dysfunction and IOP elevation. Here, we show that chronic ER stress-induced ATF4–CHOP–GADD34 signaling axis is activated in the TM of mouse and human glaucoma. Expression of ATF4 in TM increases de novo protein synthesis, ER client protein load and ER stress leading to TM cell death, IOP elevation and glaucomatous neurodegeneration in mice. Notably, ATF4 induces CHOP, which is required for ATF4-induced IOP elevation and TM cell death. ATF4 also induces GADD34, which is required for ATF4-induced protein synthesis and TM cell death. Genetic depletion or pharmacological inhibition of ATF4 rescues Dex or MYOC-induced IOP elevation most likely by preventing protein synthesis and reducing ER client protein load in TM cells. Importantly, increased de novo protein synthesis is associated with induction of ATF4–CHOP–GADD34 pathway in human primary TM cells from POAG donor eyes. These studies indicate a pathological role of ATF4 in glaucoma by promoting aberrant protein synthesis in TM cells.

Although the chronic ER stress pathway is implicated in several neurodegenerative diseases, its role in glaucoma is not known[77–79]. In the present study, we have shown that ATF4, CHOP, and GADD34 levels are increased in human primary TM cells and tissues from POAG donor eyes. In addition, we have previously shown that CHOP is increased in human glaucomatous TM tissues[36]. Of note, several of glaucoma donors are likely to be treated with IOP lowering drugs, which may positively or negatively alter ER stress markers. Since similar induction of ATF4, CHOP, and GADD34 was also associated with IOP elevation in mouse model of Dex-induced ocular hypertension and in human primary TM

cells from glaucoma donors, it is likely that induction of chronic ER stress is truly a key feature of glaucomatous TM pathology. ATF4 and CHOP can also be activated by integrated stress response (ISR) pathway[80]. In our models, induction of ATF4, CHOP, and GADD34 was associated with classical markers of ER stress including spliced XBP-1 and increased GRP78 and GRP94. These studies establish that chronic ER stress is the underlying key cause of ATF4–CHOP–GADD34 signaling pathway activation. To our knowledge, this is the first study that links activation of ATF4–CHOP–GADD34 signaling axis to the pathophysiology of POAG.

Forced expression of ATF4 in the TM exhibits many features of POAG including an open iridocorneal angle, reduced aqueous humor outflow and sustained IOP elevation and leads to glaucomatous neurodegeneration. In addition, expression of ATF4 increases protein synthesis and ER client protein load, leading to TM cell death. Loss of TM cellularity is observed in TM tissues of POAG donor eyes[15]. However, the exact cause of TM cell death is not understood. Based on our findings that ATF4 is induced in human glaucomatous TM tissues and expression of ATF4 leads to TM cell death, it is possible that induction of ATF4 in TM is associated with loss of TM in POAG. It has long been speculated that a loss of TM cells would lead to denudation of collagen structures and/or resulting in the collapse of the beams within the TM tissue. Such a collapse in the beams would decrease intertrabecular spaces thus restricting aqueous outflow and increase in IOP. Subsequently, these structural changes are likely to alter TM mechanics although this effect would be a secondary effect and not the direct causation of IOP increase. In addition to ocular hypertension, the presence of functional and structural loss of RGCs, as well as optic nerve degeneration makes this an attractive inducible mouse model to study the pathophysiology of glaucomatous TM damage and neurodegeneration.

Although adenoviral expression of ATF4 led to sustained and significant IOP elevation, it was difficult to assess glaucomatous neurodegeneration due to ocular inflammation (Supplementary Fig. 7). Ocular inflammation was observed in both Ad5.Empty and ATF4-injected mice further suggesting an immune response to adenovirus, and not the result of expression of ATF4. Previous studies have shown that Ad5 intravitreal injections can lead to mild to moderate ocular inflammation, although this inflammation does not affect IOP measurements[81]. Since the presence of ocular inflammation limited our ability to study RGC loss due to the presence of inflammatory cells in the retina and it is known that lentiviral particles cause minimal ocular toxicity[58], we repeated these experiments using lentivirus to express ATF4. Expression of ATF4 led to sustained and significant IOP elevation without ocular inflammation. We also demonstrated that sustained IOP elevation without ocular inflammation causes glaucomatous neurodegeneration as evident from significant functional and structural RGC loss and optic nerve degeneration within 10 weeks of injection.

In our previous study, we have shown that the downstream chronic ER stress transcriptional factor, CHOP, is also induced in the TM of POAG donor eyes[36]. It is known that induction of CHOP is associated with cell death and pathology in several diseases[82]. It is likely that increased expression of CHOP in TM is involved in key glaucoma pathology, but expression of CHOP alone is not sufficient to cause these pathologies since it lacks other interacting factors of the chronic ER stress pathway including ATF4. This is supported by several observations in our study. First, ATF4 is upstream of CHOP and ATF4 induces CHOP expression, while CHOP does not induce ATF4 expression. Second, a strong interaction between ATF4 and CHOP is observed only in Ad5-ATF4-transduced TM cells but not in CHOP-transduced TM cells. Therefore, in the absence of a

feedback loop, expression of CHOP does not increase ATF4, and thus sufficient interaction between ATF4 and CHOP is not possible in Ad5.CHOP-injected mice. Third, expression of ATF4 does not elevate IOP in CHOP knockout mice (in which ATF4 and CHOP interaction is obviously not possible) clearly indicating the necessity of ATF4 and CHOP physical interaction to induce IOP elevation. Our work is supported by a previous study that demonstrated that ATF4 interacts with CHOP to induce target gene expression and cell death[51]. The relationship between ATF4 and CHOP have been previously studied[51,83], and different from the findings of our study is that CHOP alone can lead to cell death. More recent studies suggest that CHOP requires ATF4 to induce cell death[51]. Based on these studies and our findings, we propose that the effects of CHOP alone are tissue dependent. While CHOP alone may be sufficient to induce cell death in neurons, CHOP requires ATF4 to induce cell death in TM cells.

Our study points out that ATF4 promotes protein synthesis and ER client protein load, which can result in TM dysfunction and cell death leading to IOP elevation. This is evident from increased de novo protein accumulation in the ER, which is reduced after ATF4 depletion. In addition, abnormal accumulation of WT MYOC (normally secreted) in ATF4-expressing cells suggest that ATF4 leads to increased ER client protein load, which can further contribute to dysfunction of the ER leading to ER stress and cell death. Consistent with this, ATF4 leads to induction of ER stress in TM cells in vitro and in vivo. Expression of ATF4 led to increased intracellular accumulation of triton insoluble form of WT and mutant MYOC. We also observed abnormal accumulation of ECM proteins in ATF4-transduced human TM cells and mouse TM tissues. Marciniak et al. demonstrated that expression of CHOP induces cell death by increasing protein synthesis and ER protein client load[84]. However, we did not observe similar effects of CHOP in our experimental settings. It is thus likely that effects of CHOP alone in regulating ER client protein load are tissue specific and in the case of TM, CHOP requires ATF4 to induce such changes. Nonetheless, our study clearly demonstrates the role of ATF4 in increasing ER protein client load, which is involved in glaucomatous TM pathology. Consistent with this, we observed that both genetic and pharmacological knockdown of the ATF4 pathway rescues mouse models of glaucoma. Furthermore, ATF4 depletion reduces Dex-induced ER protein load and prevents fibrosis of the TM.

It is interesting to note that inhibition of ATF4 rescues both Dex and mutant MYOC-induced ocular hypertension. We have previously shown that CHOP is induced in $Tg\text{-}MYOC^{Y437H}$ mice and induction of CHOP is associated with TM cell death[38]. It is therefore possible that the ATF4–CHOP pathway is involved in aggravating MYOC misfolding and depletion of ATF4 can maintain ER homeostasis and reduce misfolded MYOC by directly inhibiting MYOC protein synthesis. Future studies will be aimed at understanding the role of ATF4 in MYOC misfolding.

It is intriguing that ISRIB reduced Dex-induced protein synthesis despite several studies have shown that ISRIB increases protein synthesis by enhancing activity of elF2B[73,85,86]. Since ISRIB blocks downstream effects of phosphorylated elF2α, it can inhibit ATF4–CHOP–GADD34 signaling and also restore general protein synthesis. We propose that ISRIB's inhibitory effect on Dex-induced protein synthesis is primarily driven by its ability to effectively block ATF4–CHOP–GADD34 signaling in TM cells. ISRIB's inhibitory effect on GADD34 would result in reduced phosphatase to inhibit phosphorylation of eIF2α, which will further reduce protein synthesis. In support of this, we clearly demonstrate that treatment with ISRIB significantly reduces Dex-induced protein synthesis in TM cells, which is associated with a strong reduction in GADD34, ATF4, and CHOP levels while its

effects on phosphorylation of eIF2α are minimal. A recent study by Wang et al. (2019) independently demonstrated that the eIF2α dephosphorylation inhibitor, Salubrinal prevents tunicamycin-induced TM cell death[87]. Moreover, dominant negative inhibitor of ATF4 reduced Dex-induced protein synthesis and ocular hypertension. Alternatively, there may be another pathway by which GADD34 directly regulates protein synthesis. It is also likely that ISRIB has differential effects on chronic ER stress compared to ISR since most of effects of ISRIB are studied in context of the ISR pathway. In line with this, another study by Rabouw et al. (2019) demonstrated that ISRIB promotes protein synthesis when p-eIF2α levels are low but ISRIB is ineffective when p-eIF2α levels are high (in context of ISR)[74].

In summary, our studies indicate that ATF4–CHOP–GADD34 signaling pathway is induced in glaucoma, which promotes protein synthesis and ER client protein load, inducing TM cell death and IOP elevation. ATF4–CHOP–GADD34 signaling axis plays a pathological role in TM dysfunction leading to IOP elevation and inhibition of this pathway represents an attractive therapeutic target for slowing the progression of this disease.

## Methods

**Antibodies and reagents**. Antibodies were purchased from the following sources; fibronectin (catalog # Ab2413, Abcam), KDEL (catalog # Ab12223, Abcam), collagen I (catalog # NB600-408, Novus Biologicals), ATF4 (catalog # SC-200, Santa Cruz Biotechnology), CHOP (catalog # 13172, Novus Biologicals), GRP78 (catalog # ab21685, Abcam), GRP94 (catalog # 11402, Santa Cruz Biotechnology), Caspase 3 (catalog # 9662, Cell signaling technology), Cleaved PARP (catalog # 9541, Cell signaling technology), puromycin (catalog #A11138, Gibco, Life Technology), RBPMS (catalog # 118619, Gene Tex), cleaved caspase 3 (catalog # 9661, Cell signaling technology), GAPDH (catalog # 3683, Cell signaling technology), and β-Actin (catalog # 4970, Cell signaling technology, Danvers, MA, USA). For immunoprecipitation studies, a different CHOP (sc-7351, Santa Cruz Biotechnology) was used. All adenoviral vectors used in this study were obtained from ViraQuest Inc. (North Liberty, IA, USA). Lentiviral vectors expressing ATF4 or CHOP were obtained from VectorBuilder Inc.

**Experimental animals**. 2 to 3-month-old C57BL/6J mice were utilized in this study. Animals were purchased from the Jackson Laboratory (Bar Harbor, ME, USA). Animals were allowed to roam freely in their cages and had access to food (standard mouse chow) and water ad libitum and were maintained under 12 h light/12dark conditions. All experimental procedures were conducted in accordance with and adherence to the ARVO Statement for the Use of Animals in Ophthalmic and Vision Research. Experimental protocol was approved by the Institutional Animal Care and Use Committee (IACUC) of the University of North Texas Health Science Center (UNTHSC) (Protocol #: IACUC-2018-0032). Chop knock out mice were obtained from the Jackson labs.

**Viral vector injections**. Adenovirus expressing empty, GFP, CHOP, ATF4WT, and ATF4ΔRK under the control of CMV promoter were purchased from Vira-Quest[66]. Lentiviral particles expressing GFP, CHOP, and ATF4 were purchased from VectorBuilder. Intravitreal injections of viral particles were performed[41]. Anaesthetized mice were injected with Ad5 expressing either empty or CHOP or ATF4 ($2 \times 10^7$ pfu/eye) in vitreous chamber through sclera using a Hamilton syringe fitted with a sterile 33-gauge needle. Similar to Ad5, high titer ultra-pure lentiviral particles expressing either GFP or ATF4 WT or CHOP ($2 \times 10^6$ TU/eye) under the control of CBh promoter were injected intravitreally in both eyes.

**Cell culture**. Age-matched primary human TM cells (between passages 5 and 7) from normal and POAG donor eyes (four strains each) and transformed GTM3 or NTM5 cells were cultured in DMEM supplemented with 10% FBS (Hyclone Laboratories), 2 mM L-glutamine, penicillin (10,000 units/ml), and streptomycin (10 μg/ml) (Gibco BRL). Cells were maintained in a humidified incubator at 5% $CO_2$ and 37 °C. Adenoviral transductions were performed using antibiotic-free medium at 100 MOI for 24 h. Cells were then transferred to fresh medium and maintained for another 48 h. Cell lysates were collected in 1× lysis buffer containing: HEPES (50 mM), KCl (200 mM), EDTA (2 mM), $MgCl_2$ (1 mM), Triton X-100 (0.5%), glycerol (10%), 0.5 mM dithiothreitol, and protease inhibitor cocktail tablets (Roche Life Sciences, Indianapolis, IN, USA). For Dex treatment, cells were transduced with appropriate viral vectors for 24 h and treated with vehicle or Dex (100 nM) for another 3 days[88]. GTM3 cells stably expressing DsRed-tagged WT and mutant (Y437H) MYOC were generated previously[76]. These cells were treated with ISRIB or plasmids expressing Empty or ATF4 to examine effects on MYOC

intracellular accumulation. To examine whether CHX inhibits ATF4-induced protein synthesis, GTM3 cells transduced with ATF4 or CHOP were incubated with CHX (10 μg/ml) for 16 h and cell lysates were subjected to further analysis. To examine whether reduction of ATF4-induced protein synthesis prevents TM cell death, GTM3 cells transduced with ATF4 were incubated with CHX (2.5 μg/ml) for 12 h before harvesting cellular lysates. CRISPR-Cas9 targeting CHOP or GADD34 were generated[41]. GTM3 cells were first transfected with plasmids expressing CRISPR-Cas9 targeting CHOP or GADD34 for 24 h and then transduced with plasmids or adenovirus expressing ATF4 for 36 h.

**Intraocular pressure (IOP) measurement**. IOP measurements were carried out using TonoLab® rebound tonometer (Colonial Medical Supply, Franconia, NH, USA)[39,89]. All IOPs were measured during the day between 10 a.m. and 2 p.m. in a masked manner. Six individual IOP measurements were averaged to obtain the final IOP value for each eye at each time point. For measuring night-time IOPs, mice were kept in dark for 7 h (3–10 p.m.) and IOPs were measured in the dark using dim red lights[38].

**Aqueous outflow facility (C)**. Aqueous humor outflow facility (C) was established by constant flow infusion following our previously published methodology[54,55]. In brief, animals were anesthetized by intraperitoneal (i.p.) injection of a cocktail of ketamine (10 mg/ml) and xylazine (1 mg/ml) given in a volume of 10 ml/kg for induction. Both eyes then received a drop of proparacaine HCl (0.5%) for topical anesthesia. The anterior chamber was cannulated with a 30 G steel needle connected to a previously calibrated (sphygmomanometer, Diagnostix 700, Hauppage, NY, USA) BLPR-2 flow-through blood pressure transducer (World Precision Instruments (WPI), Sarasota, FL, USA) for the continuous determination of pressure within the eye. A topical ocular drop of sterile phosphate buffered saline (PBS, Amresco®, Solon, OH, USA) was also applied to each eye to prevent corneal drying. The opposing end of each transducer was connected to a three-way valve, which in turn was connected to: (1) 50 μl glass microsyringe (Hamilton Company, Reno, NV, USA) filled with sterile PBS (previously filtered through a 0.2 μm HT Tuffryn Membrane Acrodisc syringe filter; PALL Gelman Laboratory, Port Washington, NY, USA) loaded into SP101i microdialysis infusion pump with half-stepping capability (WPI), and (2) an open-ended, variable-height manometer. Signals from the pressure transducers were passed via a TBM4M Bridge Amplifier (gain setting: 1000×, WPI) and a Lab-Trax analog-to-digital converter (WPI) to a computer. Data were recorded using Lab-Scribe2 software (WPI).

The manometer was opened to the circuit and the pressure adjusted to equal the pre-cannulation IOP. Following a period of ~1 min, the manometer was closed to the circuit and eyes were infused at a flow rate of 100 nL/min. When pressure had stabilized (typically within 10–30 min), pressure measurements were recorded every 5 min for the next 15 min. Flow rates were then increased sequentially (200, 300, 400, and 500 nL/min). Three stabilized pressures (spaced 5 min apart) were recorded at each flow rate. C in each eye of each animal was calculated as the reciprocal of a slope of a plot of mean stabilized pressure as ordinate against flow rate as abscissa.

**Mouse slit-lamp examination**. Cornea and anterior chamber phenotypes were examined with a slit-lamp (SL-D7; Topcon) and photo-documented with a digital camera (D100; Nikon)[38].

**Pattern ERG**. RGC function was analyzed using binocular snout pattern ERG (PERG animal research system, JORVEC Corp., Miami, FL, USA)[59]. In brief, mice were anesthetized using intraperitoneal injection of ketamine/xylazine mixture (100, 10 mg/kg, respectively). Anesthetized mice were placed on a temperature controllable metal base at a fixed distance (10 cm) from the LED monitors and maintained at constant body temperature (37 °C) using a rectal probe. A small amount of hypromellose eye drops were applied topically to prevent corneal dryness while recording. The PERG was derived simultaneously from each eye using subcutaneous electrodes placed at snout (active), back of the head (reference) and tail (ground) in response to contrast reversal of gratings generated from two LED screens, operated at a slightly different frequency. Two consecutive repetitions were averaged and amplitudes (P1–N2) representing RGC function was graphically shown.

**RGC staining and counting**. Mice were euthanized, and enucleated eyes were fixed in 4% PFA overnight at 4 °C. Next day, eyes were washed thoroughly with 1xPBS and dissected into anterior and posterior cups. Whole retina was removed from the posterior cups, incubated in blocking buffer (10% goat serum containing 1% Triton-X-100) for 3 h. Retina was then incubated with primary antibody (Rabbit anti-RBPMS, 1:200) in a blocking buffer for 48 h at 4 °C. Retina was washed thrice with 1×PBS containing 0.1% Triton-X-100 for 30 min each. Retina was incubated with appropriate secondary antibody (Alexa flour anti-rabbit 568, 1:500) in 1×PBS for 2 h. Retina was washed thrice with 1×PBS for 30 min each and flat-mounted on a glass slide by cutting four equal quadrants attached at center of the retina. Images were taken using Keyence microscope at ×200 magnification. Total of 20 non-overlapping images per the retina including 12 from periphery (4 from each quadrant), 4 from mid-periphery (1 from each quadrant), and 4 from central

region (1 from each quadrant) were taken and used for counting of RBPMS-positive RGCs using Image J software. The total number of RGCs in peripheral, mid-peripheral, and central retina were quantified separately in each group and represented graphically.

**PPD staining of optic nerve**. Axonal degeneration in the optic nerve was examined using a PPD staining and total mean optic nerve axons were counted[38]. Briefly, optic nerves were collected from GFP and ATF4-injected mice and fixed overnight in phosphate buffer solution containing 3% glutaraldehyde/paraformaldehyde at 4 °C. The optic nerves rinsed twice for 10 min with 0.1 M phosphate buffer and once with 0.1 M sodium acetate buffer, then dehydrated in graded ethanol concentrations. After embedding in the resin, the transverse semithin sections (1 μm) were cut and stained with 1% PPD for 10 min. Total 10 images without overlap were taken using brightfield microscope (Leica DM 2000 LED) with a total amplification of ×1000. The surviving axons were counted in area of 625 square micrometers from each image and sum of total surviving axons counted approximately equals to 10% of the total optic nerve cross-sectional area.

**TUNEL assay**. TUNEL assays were performed on TM cells and mouse eye sections using a TUNEL Apo-Green Detection Kit (Biotool, Houston, USA) in accordance with the manufacturer's recommended protocol. Two weeks after Ad5.Empty or Ad5.ATF4 injections ($2 \times 10^7$ pfu/eye), animals were sacrificed and eyes were enucleated, fixed, processed, and embedded in paraffin. Five-micron sections were cut using a microtome. Sections were dewaxed, rehydrated, and permeabilized by incubating with 20 μg/ml proteinase K solution for 20 min at room temperature. GTM3 or NTM5 cells transduced with Ad5.Empty or Ad5.ATF4 (100 MOI) for 36 h and were fixed in freshly prepared 4% PFA and permeabilized using 0.2% triton-X-100 for 10 min. Both mouse eye sections and fixed TM cells were incubated with 1× equilibration buffer for 10 min and allowed 60 min incubation with the TUNEL reaction mixture (containing Apogreen labeling mix and Recombinant TdT enzyme) at 37 °C in the dark. Following three washes in PBS, sections were mounted in DAPI-mounting solution. TUNEL-positive green fluorescence images were taken using a Keyence microscope (Itasca, IL, USA).

**Immunostaining**. Enucleated eyes were fixed in 4% paraformaldehyde for 3 h. The fixed eyes were processed and embedded in paraffin. Five-micron sections were cut and utilized for immunostaining. Briefly, sections were deparaffinized and rehydrated, then subjected to antigen retrieval with citrate buffer (pH = 6). Slides were incubated with blocking buffer (10% goat serum and 0.5% Triton-X-100 in PBS) for 2 h, and then with primary antibody in blocking buffer overnight. Slides were washed three times with PBS, and then incubated for 2 h with approximate Alexa Fluor secondary antibodies (Life technologies, Grand Island, NY, USA). After three final washes in PBS, sections were mounted with DAPI-mounting solution. Images were captured using a Keyence microscope (Itasca, IL, USA). For immunostaining of human donor eyes, age-matched normal ($n = 9$) and glaucomatous ($n = 11$) anterior segment sections were stained with ATF4 and GADD34[90]. The study design and conduct complied with all relevant regulations regarding the use of human study participants and was conducted in accordance to the criteria set by the Declaration of Helsinki. Briefly, post-mortem fixed human eyes were received from the Lions Eye Institute for Transplant & Research (Tampa, FL). The Lions Eye Institute is regulated directly by the Food and Drug Administration (FDA) and it is exempted from oversight of an Institutional Review Board (IRB), since it does not conduct biomedical research involving living human research subjects. These tissues are self-donated to the Lions Eye Institute for research and Lions Eye Institute receives informed consent and complies with the Florida Agency for Healthcare Administration (AHCA) statute 59A-1 before collecting tissues.

**Co-immunoprecipitation**. GTM3 cells were transduced with Ad5-Empty, Ad5-CHOP, and Ad5-ATF4 at 100 MOI for 24 h. Co-immunoprecipitation was carried out with anti-CHOP antibody using the Dynabeads Co-Immunoprecipitation Kit (Life technologies, Grand Island, NY, USA). Briefly, CHOP and IgG antibodies (2 μg) were diluted in 200 μl of lysis buffer (used as binding and washing buffer) and incubated with 50 μl of dynabeads protein G for 10 min at room temperature on a rotating shaker. Bead–Ab complexes were washed three times using the magnet separator provided with the kit. Total protein lysates (200 μg) were incubated with bead–Ab complexes for 10 min. Beads–Ab–Ag complexes were washed three times with lysis buffer containing 0.5% Triton-X-100. The bead–Ab–Ag complexes were then dissolved in 30 μl of LDS sample buffer and heated for 10 min at 70 °C. Beads were separated using the magnet separator, and supernatants were subjected to Western blot analysis.

**Western blot analysis**. Approximately 20–30 μg total protein from either cell lysates or anterior segment lysates were run on denaturing 4–12% gradient polyacrylamide ready-made gels (NuPAGE Bis–Tris gels, Life technologies, Grand Island, NY, USA) and then transferred onto PVDF membranes. Blots were blocked with 10% non-fat dried milk for 1 h and then incubated overnight with specific primary antibodies at 4 °C on a rotating shaker. Membranes were washed thrice with PBST and incubated with corresponding HRP-conjugated secondary antibody for 90 min. Proteins were then visualized using ECL detection reagents

(SuperSignal West Femto Maximum Sensitivity Substrate; Life technologies) using LI-COR Odyssey Fc image system.

**Total ER isolation**. Total ER fractions were isolated from GTM cells using ER-enrichment kit (cat # NBP2-29482, Novus Biologicals, LLC). GTM3 cells were grown on 10 cm dishes and treated with plasmids expressing ATF4. Cells were lysed using 2 ml isosmotic homogenization buffer containing protease inhibitor cocktail. Lysates were centrifuged at $1000 \times g$ for 10 min at 4 °C. The pellet containing nuclei and cell debris was discarded and supernatant was centrifuged at $12,000 \times g$ for 15 min at 4 °C. The pellet (mitochondria and cell debris) was discarded and supernatant was ultra-centrifuged at $90,000 \times g$ (Beckman ultra-centrifuge) for 60 min at 4 °C. The supernatant was discarded and pellet containing the total ER fraction was resuspended in 1× suspension buffer containing protease inhibitor cocktail and was subjected to Western blot analysis.

**Surface sensing of translation (SUnSET) assay**. The SUnSET assay was carried out as described previously[37,91]. Primary human TM cells from normal and glaucoma donor eyes or GTM3 cells were plated in 10 cm tissue culture dishes and treated with vehicle (0.1% ethanol), Dex (100 nM), and Dex plus Ad5.ATF4ΔRK for 36 h or transfected with plasmids expressing ATF4 or CHOP for 48 h. TM cells were then incubated with puromycin (10 μg/ml) for 30 min before harvesting cell lysates. Cells were washed thrice with 1 × PBS and used for the isolation of total ER fractions. To quantify the amount of puromycin incorporated in the newly synthesized proteins in the ER, the ER lysates were subjected to Western blot analysis using mouse anti-puromycin primary antibody (1:10,000 dilution) and goat anti rabbit-HRP secondary antibody. GAPDH was used as a loading control.

**Reporting summary**. Further information on research design is available in the Nature Research Reporting Summary linked to this article.

## Data availability
All data generated and analyzed during this study are included in this published article (and its supplementary information files). Source data are provided with this paper.

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

## Acknowledgements
The authors would like to thank Dr. Abbot Clark, Dr. Colleen McDowell, Sherri Harris, and Sandra Maansson for assistance with some experiments. These studies were supported by the National Institutes of Health (EY028616 and EY026177) and bright focus glaucoma foundation.

## Author contributions
R.B.K. and G.S.Z. designed research studies, analyzed data, provided reagents and wrote the manuscript. P.M., P.D.P., C.C.S., J.C.M. V.R. and V.C.S. assisted in conducting key experiments, provided reagents, and assisted in manuscript preparation. All authors discussed the results and implications and commented on the manuscript at all stages.

## Competing interests
The authors declare no competing interests.

## Additional information



