## [Peer Review File · Nature Communications]

Reviewers' comments:

Reviewer #1 (Remarks to the Author):

The authors have a history of examining ER stress in the context of MYOC (Y437H) and steroid induced glaucoma. The current work adds to this story by clarifying the role of ATF4, a known player in ER stress and UPR, and its interactions with CHOP that are necessary to lead to outflow dysfunction in some mouse models of glaucoma.

The evidence for ATF4 overexpression in glaucomatous human TM is limited to an immunofluorescence micrograph. The images in Fig 1D seem to show brighter ATF4 labelling in the glaucomatous TM compared to the normal, but IF is nonquantitative and sensitive to protocol and image settings. When biochemistry is used to examine ATF4 expression in glaucomatous TM cells, the expression appears non-existent (Fig 4a,b), but this is not directly compared against normal TM cells. Perhaps the loss of ATF4 expression between tissue and culture is an in vitro artefact, or perhaps ATF4 was never elevated to begin with. The authors refer to previous work (Peters et al., IOVS 2015) to claim elevated ATF4 expression in glaucoma, but elevated ATF4 expression in steroid-induced glaucoma is hardly a surprise given the prior report of steroid induction of ER stress and the known role of ATF4 in ER stress.

ER stress is sensitive to protein load. Thus, viral induction could potentially lead to ER stress by overexpressing misfolded proteins that overload the ER. Some viruses (human and mouse CMV; PMID: 22496230, 15890928) even induce ER stress. In this view, is Ad5-Empty that has no product really an appropriate control? How do we know that the results are not a consequence of protein overexpression per se? No details are given on the promoters. If the promoters are strong (CMV, SV40) one could question whether such high expression is representative of the glaucomatous state. The authors may know that overexpression has been identified as a potential artefact causing ER stress in mouse models of Alzheimer's (for example, Hashimoto et al, JBC, 2018; PMID 29298895). Perhaps something similar is happening in mouse models of glaucoma.

Biochemistry seems to have been conducted on only one cell line (GTM3) with n=2 appearing to refer to technical (not biological) replicates. Using only one cell line is risky when we know that there is quite a bit of diversity between human donors. The senior author was a co-author on "Consensus recommendations for trabecular meshwork cell isolation, characterisation and culture" (Keller et al., Exp Eye Res 2018), which recommends "Ideally, three or more independent biological replicates with studies performed in triplicate are needed in order to account for the diverse genetic backgrounds of humans."

The authors should be aware that there is limited ultrastructural evidence for fibrosis in the TM of POAG eyes. Despite many decades of investigation, the strongest evidence for ECM accumulation in the TM is the report of increased plaque-like sheath material (reviewed by Tektas and Lütjen-Drecoll, Exp Eye Res, 2009). Morphometric analysis showed that this material is likely insufficient to explain the glaucomatous elevation in outflow resistance (Murphy et al., Arch Ophthalmol 1992). Other ultrastructural studies have shown ECM deposits in the TM of humans and mice with steroid induced glaucoma, but these changes are subtle and likely insufficient to explain the increased outflow resistance (Johnson et al., Arch Ophthalmol 1997). No ultrastructural evidence is given in this manuscript (or in previous manuscripts: Zode et al., JCI 2011/14, Peters et al., IOVS 2015, Kasetti et al Sci Rep 2017, Patel et al., Am J Path 2017) to support the claims of true fibrosis (ie ECM accumulation) as the cause of decreased outflow resistance. IF images of fibronectin are used to support the fibrosis argument, but it is not clear whether labelling is intracellular or extracellular, although colocalization with ATF4 suggests intracellular. The link between ER stress and true fibrosis therefore seems quite thin.

IOP elevations and decrease in facility confirm previous work that ER stress induced by DEX treatment or Y437H are associated with elevated IOP and reduced outflow facility (Patel et al. Am J

Path 2017; Zode et al., JCI 2014).

Reviewer #2 (Remarks to the Author):

This manuscript addresses the pathological mechanisms for primary open-angle glaucoma (POAG). Previous reports indicated that endoplasmic reticulum (ER) stress and the ensuing Unfolded protein response (UPR) can be a contributor to damage to glaucomatous trabecular meshwork (TM) and elevated intraocular pressure (IOP). The UPR features three sensors of ER stress, which trigger translational and transcriptional programs of gene expression designed to expand the processing capacity of the ER. This manuscript addresses the model that ATF4 and CHOP, two transcription factors induced by the UPR, are contributors to the pathology underlying POAG. ATF4 and CHOP induce the transcription of hundreds of UPR genes. While the UPR has a primary adaptive function, chronic induction of CHOP expression can instead result in cell death processes. For example, ATF4 and CHOP induce the expression of GADD34, which participates in the feedback dephosphorylation of eIF2, which facilitates translation of genes in the UPR transcriptome. However, premature resumption of translation during chronic ER stress can have detrimental consequences on cell health (ref 50 and 75). Using in vitro and in vivo models, the manuscript concludes that induction of ATF4 is paramount to the pathologies of POAG. The central conclusions would be of broad interest and would be consistent with UPR concepts in other organ pathologies. The experiments adopt appropriate models, but the UPR concepts are less well developed, too simplified, and likely inconsistent with the model that ATF4 is detrimental because it facilitates resumption of translation. Overall, the manuscript has significance to the field, but needs further development to support the stated conclusions and clarify some concepts that are opaque in the manuscript. Below are some broad concerns.

Reviewer concerns:

1. The lead-in model system involves TM tissues from a glaucoma model model and human POAG donor eyes. It is stated that there may be ER stress involving Dex-induced hypertension, but this is associated with largely ATF4 and CHOP expression, which can also be induced by cytosolic stresses (ISR). It is noted that prior publications support in part this idea, but it is important to shore up the UPR conclusions in this manuscript beyond GRP expression (e.g. XBP1s). The underlying chronic ER stress is a key to the ATF4/translation regulation model and this needs to be shown with the appropriate stress timeline.
2. The manuscript is problematic with the adenoviral delivery of ATF4 and CHOP. ATF4 overexpression appears to induce IOP and glaucoma, whereas CHOP does not. Both ATF4 and CHOP are basic zipper transcription factors, so in the apparent absence of stress one does not know if the UPR-programmed partners are associated. Therefore, the CHOP conclusions should be cautiously considered. For the same dimerization ideas, one should be cautious about interpreting the phenotype of expression of the dominant-negative version of ATF4 (multiple binding partners). It is likely that CHOP is involved in the key pathologies, but the availability of the appropriate binding partners is critical for function. Here it would be helpful to incorporate some genome-wide gene expression data into the manuscript to assess function with the ATF4 and CHOP expression.
3. The data for the protein synthesis changes in Fig. 5A was not convincing. The model that induced ATF4 expression enhances protein synthesis and this is detrimental for TM and tissues was not convincing. This is the heart of the proposed model in the manuscript and one would need to incorporate additional markers and controls to sufficiently support the stated model. For example, more convincing protein synthesis measurements, measurements of total and phosphorylated eIF2 and GADD34, and consider an artificial block in translation (e.g. modest amounts of cycloheximide or the like) to address whether lowered translation restores health.
4. The manuscript proposes that the reason why ATF4 is deleterious to TM is that the transcription

factor (associated with CHOP or other partner) triggers increased translation (through GADD34?) and elevated ER client load (chronic UPR stress). The manuscript later indicates that ISRIB lowered Dex-treated IOP and TM fibrosis. ISRIB would be expected to thwart translational control by PERK induced eIF2-P and as a consequence block ATF4 and CHOP expression. However, ISRIB would also be expected to allow for high levels of mRNA translation during Dex-induced ER stress. If the model is that ATF4 is deleterious to cells because it allows for increased translation and overload of ER, would not ISRIB also drive high levels of protein synthesis during the Dex-induced ER stress? This would appear to be inconsistent with the model. Perhaps ATF4/CHOP target genes involved in cell death pathologies (DR5 and the like) are critical to these pathologies?

5. The increase in cellular ROS due to overexpression in ATF4 is confusing. One often views ATF4 as having antioxidation properties (see Harding 2003 Mol Cell for lead-in). It is noted that ATF4 is tied to mitochondrial stress adaptation and resumption of translation can affect influx of proteins into the ER and disulfide bond formation, which can affect ROS.

Authors Response Letter

Dear Editor and Reviewers,

We greatly appreciate the reviewers' valuable insights and suggestions, which have substantially improved our manuscript. We have carefully addressed all concerns. Changes and new experiments added to the manuscript are underlined in the revised manuscript and are also summarized below.

- 1) In the revised manuscript, we provide strong evidence to support that increased protein synthesis is a key feature of glaucomatous TM cells, which leads to TM dysfunction and death, elevating IOP. We demonstrate that ATF4-CHOP-GADD34 signaling is activated in glaucomatous TM cells/tissues, which is responsible for promoting protein synthesis, leading to TM dysfunction and IOP elevation. Specifically, knockdown of ATF4, CHOP or GADD34 individually is sufficient to reduce protein synthesis and rescue glaucomatous phenotypes. This directly addresses the reviewer's concern regarding novelty over previous work related to PBA.
- 2) Human primary TM cells from glaucomatous donor eyes (n=4 cell strains) demonstrated significantly increase protein synthesis, which is associated with increased GADD34, XBP-1, ATF4 and CHOP levels compared to TM cells cultured from normal donor eyes.
- 3) ATF4 and GADD34 levels are significantly increased in human TM tissues from POAG donor eyes compared to age-matched normal donor eyes.
- 4) Increased levels of GRP78, spliced XBP-1, ATF4, GADD34 and CHOP levels in the TM tissues are associated with Dex-induced ocular hypertension. These data demonstrate that ATF4 and CHOP induction in our model is due to chronic ER stress and not due to activation of the integrated stress response pathway (ISR).
- 5) ATF4 increases de novo protein synthesis and TM cell death, which is associated with induction of CHOP and GADD34. Inhibition of general protein synthesis by cyclohexamide reduced ATF4-induced TM cell death. Moreover, ATF4-induced protein synthesis and TM cell death is dependant on GADD34 and CHOP, as knockdown of GADD34 or CHOP reduces both protein synthesis and prevents TM cell death.
- 6) ISRIB reduced Dex-induced protein synthesis, which was associated with inhibition of ATF4-CHOP-GADD34 protein levels. ISRIB-mediated reduction of protein synthesis is likely due to chronic inhibition of ATF4-CHOP-GADD34 signaling, while its effects on phosphorylation of the eukaryotic translation initiation factor (eIF2 α) may be transient.
- 7) All experiments were repeated a minimum of three times and statistical analysis was included whenever possible. In addition, we have now included human primary TM cells from normal (n=4 cell strains) and POAG (n=4 cell strains) donor eyes.

Detailed responses to the reviewers concerns are listed below:

Reviewer#1:

Reviewer comment 1) The authors have a history of examining ER stress in the context of MYOC (Y437H) and steroid induced glaucoma. The current work adds to this story by clarifying the role of ATF4, a known player in ER stress and UPR, and its interactions with CHOP that are necessary to lead to outflow dysfunction in some mouse models of glaucoma.

Authors Response: We feel strongly that the data presented in our revised manuscript are novel and make a significant advancement over previously published work in several aspects. **1)** Regarding our previous history of studying ER stress, this manuscript makes a significant advancement in the field as we demonstrate that ATF4-CHOP-GADD34 signaling is induced in POAG, which leads to trabecular meshwork dysfunction and cell death. The experimental design is based on findings that markers of chronic ER stress, ATF4, CHOP and GADD34 are significantly increased in TM cells and tissues from POAG donor eyes. Notably, we show that the pathological induction of ATF4-CHOP-GADD34 leads to increased protein synthesis and TM cell death, resulting in glaucoma. This has not been investigated prior and provides a significant advancement in the field of ER stress as pointed out by the second reviewer. **2)** In our revised manuscript, we propose a new concept that has not been investigated in the glaucoma field. We propose that aberrant protein synthesis is associated with glaucoma pathogenesis, which is caused by induction of the ATF4-CHOP-GADD34 signaling pathway in the TM. Based on this concept, we provide new avenues of treatment targeting ATF4-CHOP-GADD34 signaling as demonstrated using ISRIB or selective knockdown of ATF4-CHOP-GADD34. Note that PBA works by correcting misfolded

Figure 1: Proposed model showing the roles of ATF4-CHOP-GADD34 in glaucoma pathogenesis.

proteins and it does not have any effect on ATF4-CHOP-GADD34 signaling. While PBA works on subset of glaucoma cases, the proposed studies provide a mechanistic insights and possible treatment that is applicable to all cases of POAG. **3)** Comparing our previous work that linked the role of ER stress to the pathogenesis of MYOC (4% of POAG) or glucocorticoid-induced glaucoma (accounting for only one subset of glaucoma), our revised manuscript is aimed at understanding the role of chronic ER stress in general POAG and not just limited forms of glaucoma. This is immensely important since glaucoma is heterogeneous and multifactorial and we need to understand and develop targeted treatments that will apply to

general POAG. **4)** Our current study provides novel avenues for the treatment of POAG targeting ATF4-CHOP-GADD34 using ISRIB or CRISPR-Cas9 knockdown of ATF4/CHOP/GADD34, which was not discussed in our previous work.

Comment 2) The evidence for ATF4 overexpression in glaucomatous human TM is limited to an immunofluorescence micrograph. The images in Fig 1D seem to show brighter ATF4 labelling in the glaucomatous TM compared to the normal, but IF is nonquantitative and sensitive to protocol and image settings. When biochemistry is used to examine ATF4 expression in glaucomatous TM cells, the expression appears non-existent (Fig 4a,b), but this is not directly compared against normal TM cells. Perhaps the loss of ATF4 expression between tissue and culture is an in vitro artefact, or perhaps ATF4 was never elevated to begin with. The authors refer to previous work (Peters et al., IOVS 2015) to claim elevated ATF4 expression in glaucoma, but elevated ATF4 expression in steroid-induced glaucoma is hardly a surprise given the prior report of steroid induction of ER stress and the known role of ATF4 in ER stress.

Authors Response: We have added several new experiments to support that chronic ER stress-induced ATF4-CHOP-GADD34 signaling is activated in glaucoma (**Fig.1**). **1)** As suggested by the reviewer, we have examined these chronic ER stress markers in human primary TM cells obtained from normal and POAG donors. Western blot and densitometric analysis clearly demonstrated significantly increased ATF4, CHOP and GADD34 protein levels in glaucomatous TM cells. **2)** We have repeated immunostaining on age-matched human normal and glaucomatous anterior segment tissues along with GADD34. Intensity measurement clearly demonstrated significant induction of ATF4 and GADD34 in the TM tissues of POAG donor eyes. **3)** It should be noted that GTM3 is a transformed cell line, which is primarily utilized to express plasmids since it is difficult to transfect primary TM cells. Expression pattern of control GTM3 should not be considered as a representation of glaucomatous state. Also, ATF4, CHOP and GADD34 are induced during chronic ER stress and baseline expression is expected to be minimal without stress.

We agree that induction of ATF4 in steroid-induced glaucoma is not surprising based on our previous papers. We would like to clarify that chronic ER stress analysis (induction of ATF4) in our mouse model of Dex-induced OHT was performed to re-establish a relevant mouse model of ER stress, and was later used to target chronic ER stress using ISRIB or CRISPR-Cas9 genome editing. In the revised version, we demonstrate that ATF4-CHOP-GADD34 axis is induced in glaucomatous TM cells/tissues, which was not previously described. This is an important finding since we further demonstrate that induction of ATF4-CHOP-GADD34 signaling promotes aberrant protein synthesis leading to ER stress-mediated TM cell dysfunction and death, thereby elevating IOP. Notably, inhibition of specific ATF4-CHOP-GADD34 via ISRIB or genetic ATF4 knockdown rescued mouse models of ocular hypertension.

Comment 3) ER stress is sensitive to protein load. Thus, viral induction could potentially lead to ER stress by overexpressing misfolded proteins that overload the ER. Some viruses (human and mouse CMV; PMID: 22496230, 15890928) even induce ER stress. In this view, is Ad5-Empty that has no product really an appropriate control? How do we know that the results are not a consequence of protein overexpression per se? No details are given on the promoters. If the promoters are strong (CMV, SV40)

one could question whether such high expression is representative of the glaucomatous state. The authors may know that overexpression has been identified as a potential artefact causing ER stress in mouse models of Alzheimer's (for example, Hashimoto et al, JBC, 2018; PMID 29298895). Perhaps something similar is happening in mouse models of glaucoma.

Authors Response: We agree with the reviewer that ER stress is highly sensitive to several factors. Therefore, we include several controls and experiments in our revised manuscript to support our findings. **1)** In regards to Ad5-Empty being an inappropriate control, we now show that adenoviral expression of GFP does not significantly elevate IOP while Ad5-expressing ATF4 leads to significant IOP elevation (S.I. 6). In addition, lentiviral particles expressing GFP did not elevate IOP significantly (Fig.3A). **2)** In regards to the effect of overexpression of certain proteins/viruses on ER stress, we clearly show that lentiviral particles expressing ATF4, but not GFP or CHOP, elevate IOP significantly (Fig. 3A) and also induce ER stress in mice (Fig. 5F and Sl. 14). Similarly, adenoviral expression of ATF4 (but not CHOP) under the control of a CMV promoter elevated IOP significantly (data not shown), clearly demonstrating that ATF4-induced OHT is not due to overexpression of proteins under a strong promoter. In addition, ER stress was observed in human primary TM cells and tissues from POAG donor eyes (Fig.1) without use of any virus or treatment. Also, Dex treatment lead to chronic ER stress, which is associated with IOP elevation (Fig. 1 and Zode et al, 2014). **3)** Regarding use of strong promoters, we have now included detailed information regarding promoters in Methods and Materials. Adenoviral expression of ATF4 is driven by the CMV promoter while lentiviral expression is driven by the CBh promoter. Unfortunately, there is no specific promoter for TM tropism and several labs have utilized adenoviral expression under a CMV promoter to express the gene of interest in mouse TM. Most importantly, we show that adenoviral expression of ATF4 Δ RK, which overexpresses the dominant negative inhibitor of ATF4, reduces Dex-induced OHT and ER stress in mouse TM (Fig.7D-E and S.I. 13). The fact that expression of ATF4 but not GFP or CHOP or ATF4 Δ RK-elevated IOP and induced ER stress clearly demonstrate that glaucoma phenotypes in our model are not due to overexpression and likely due to activation of ATF4-mediated pathways in TM cells. **4)** It should be noted that key experiments showing that ATF4 promotes protein synthesis and ER stress (Fig. 5A-B) as well as TM cell death (Fig.6A-J) were performed using plasmids expressing ATF4 to rule out any effects from virus (data not shown) and similar findings were observed.

Comment 4) Biochemistry seems to have been conducted on only one cell line (GTM3) with n=2 appearing to refer to technical (not biological) replicates. Using only one cell line is risky when we know that there is quite a bit of diversity between human donors. The senior author was a co-author on "Consensus recommendations for trabecular meshwork cell isolation, characterisation and culture" (Keller et al., Exp Eye Res 2018), which recommends "Ideally, three or more independent biological replicates with studies performed in triplicate are needed in order to account for the diverse genetic backgrounds of humans."

Authors Response: We thank the reviewer for pointing out this oversight. We have

now included several experiments to address this concern. We have repeated all of our experiments in GTM3 cells to reflect n=3 and performed densitometric and statistical analysis. In addition, we have included data from NTM5 in some cases to support our key findings (i.e. ATF4 leads to TM cell death). We have also analyzed chronic ER stress markers in human primary TM cells from age-matched normal and POAG donor eyes (n=4 cell strains). It should be noted that *in vivo* studies were performed to replicate findings from cell culture studies. In addition, we have utilized primary human TM cells (n=3 cell strains) to show that ATF4 leads to ER stress in supplementary figure 13.

Comment 5) The authors should be aware that there is limited ultrastructural evidence for fibrosis in the TM of POAG eyes. Despite many decades of investigation, the strongest evidence for ECM accumulation in the TM is the report of increased plaque-like sheath material (reviewed by Tektas and Lütjen-Drecoll, *Exp Eye Res*, 2009). Morphometric analysis showed that this material is likely insufficient to explain the glaucomatous elevation in outflow resistance (Murphy et al., *Arch Ophthalmol* 1992). Other ultrastructural studies have shown ECM deposits in the TM of humans and mice with steroid induced glaucoma, but these changes are subtle and likely insufficient to explain the increased outflow resistance (Johnson et al., *Arch Ophthalmol* 1997). No ultrastructural evidence is given in this manuscript (or in previous manuscripts: Zode et al., *JCI* 2011/14, Peters et al., *IOVS* 2015, Kasetti et al *Sci Rep* 2017, Patel et al., *Am J Path* 2017) to support the claims of true fibrosis (ie ECM accumulation) as the cause of decreased outflow resistance. IF images of fibronectin are used to support the fibrosis argument, but it is not clear whether labelling is intracellular or extracellular, although colocalization with ATF4 suggests intracellular. The link between ER stress and true fibrosis therefore seems quite thin.

Authors Response: We agree with the reviewer that the link between fibrosis in glaucoma is weak and it is not clear whether changes in ECM protein are a result of upstream pathology or the cause of TM dysfunction. We are not claiming that ECM accumulation is a causative factor in glaucoma, nor ATF4-induced ECM accumulation leads to IOP elevation. There are two main reasons that we include analysis of ECM proteins. **1)** Since ECM markers including fibronectin have been shown to be increased in glaucomatous TM tissues and several conditions are known to alter ECM in TM (Medina-Ortiz et al, 2013, Kasetti et al, 2017, Ago et al, 2014, Fleenor et al, 2006, Vranka et al, 2015, Bradley et al, 2001 and Vittal et al, 2005), we utilized fibronectin and collagen protein changes as a measure of TM homeostasis. Although there is a link of chronic ER stress with ECM proteins, this is not the focus of this manuscript. **2)** As the reviewer pointed out, there appears to be intracellular accumulation of fibronectin in glaucomatous TM. We propose that ATF4 increases protein synthesis and ER client protein load. Consistent with this, we observed increased accumulation of intracellular fibronectin and WT or mutant myocilin in cells expressing ATF4 further suggesting that ATF4 increases ER client load and intracellular accumulation.

Comment 6) IOP elevations and decrease in facility confirm previous work that ER stress induced by DEX treatment or Y437H are associated with elevated IOP and reduced outflow facility (Patel et al. Am J Path 2017; Zode et al., JCI 2014).

Authors Response: We would like to emphasize that outflow facility shown in Figure 2B was performed to determine whether ATF4-induced OHT is caused by reduced outflow facility (i.e. TM dysfunction). We did not study whether Dex or mutant MYOC reduces outflow facility. It should be noted that this manuscript focused on understanding basic pathways/mechanisms of how chronic ER stress leads to TM cell death/dysfunction. Our data clearly shows that activated ATF4-CHOP-GADD34 signaling increases protein synthesis and ER client protein load, leading to TM dysfunction/cell death and IOP elevation. Our studies provide novel mechanistic insights and provide possible treatment avenues for glaucoma, targeting this pathway.

Reviewer # 2: (Remarks to the Authors)

This manuscript addresses the pathological mechanisms for primary open-angle glaucoma (POAG). Previous reports indicated that endoplasmic reticulum (ER) stress and the ensuing Unfolded protein response (UPR) can be a contributor to damage to glaucomatous trabecular meshwork (TM) and elevated intraocular pressure (IOP). The UPR features three sensors of ER stress, which trigger translational and transcriptional programs of gene expression designed to expand the processing capacity of the ER. This manuscript addresses the model that ATF4 and CHOP, two transcription factors induced by the UPR, are contributors to the pathology underlying POAG. ATF4 and CHOP induce the transcription of hundreds of UPR genes. While the UPR has a primary adaptive function, chronic induction of CHOP expression can instead result in cell death processes. For example, ATF4 and CHOP induce the expression of GADD34, which participates in the feedback dephosphorylation of eIF2, which facilitates translation

of genes in the UPR transcriptome. However, premature resumption of translation during chronic ER stress can have detrimental consequences on cell health (ref 50 and 75). Using in vitro and in vivo models, the manuscript concludes that induction of ATF4 is paramount to the pathologies of POAG. The central conclusions would be of broad interest and would be consistent with UPR concepts in other organ pathologies. The experiments adopt appropriate models, but the UPR concepts are less well developed, too simplified, and likely inconsistent with the model that ATF4 is detrimental because it facilitates resumption of translation. Overall, the manuscript has significance to the field, but needs further development to support the stated conclusions and clarify some concepts that are opaque in the manuscript. Below are some broad concerns.

Reviewers Concerns:

Comment 1) The lead-in model system involves TM tissues from a glaucoma model and human POAG donor eyes. It is stated that there may be ER stress involving Dex-induced hypertension, but this is associated with largely ATF4 and CHOP expression, which can also be induced by cytosolic stresses (ISR). It is noted that prior publications support in part this idea, but it is important to shore up the UPR conclusions in this manuscript beyond GRP expression (e.g. XBP1s). The underlying chronic ER

stress is a key to the ATF4/translation regulation model and this needs to be shown with the appropriate stress timeline.

Authors Response: We thank the reviewer for providing encouraging comments and key input to improve our manuscript. We feel that constructive suggestions from this reviewer immensely helped us to improve the quality of this manuscript. We strongly agree with the reviewer that additional markers are needed to support the role of chronic ER stress and separate our studies from ISR. We have now included analysis of XBP-1 (a key marker of UPR) in our mouse model of Dex-induced ocular hypertension and in human glaucomatous TM cells. As suggested by the reviewer, we have examined the roles of GADD34 in ATF4-induced protein synthesis. We now show that both ATF4 and GADD34 levels are increased in human primary TM cells and TM tissues from POAG donor eyes. Also, GADD34 knockdown inhibits ATF4-mediated protein synthesis and TM cell death. Therefore, it is likely that ATF4 effects on protein synthesis and cell death are due to induction of GADD34, which leads to dephosphorylation of eIF2 α and recovery of protein synthesis.

Comment 2) The manuscript is problematic with the adenoviral delivery of ATF4 and CHOP. ATF4 overexpression appears to induce IOP and glaucoma, whereas CHOP does not. Both ATF4 and CHOP are basic zipper transcription factors, so in the apparent absence of stress one does not know if the UPR-programmed partners are associated. Therefore, the CHOP conclusions should be cautiously considered. For the same dimerization ideas, one should be cautious about interpreting the phenotype of expression of the dominant-negative version of ATF4 (multiple binding partners). It is likely that CHOP is involved in the key pathologies, but the availability of the appropriate binding partners is critical for function. Here it would be helpful to incorporate some genome-wide gene expression data into the manuscript to assess function with the ATF4 and CHOP expression.

Authors Response: We agree with the reviewer that CHOP is involved in the key glaucoma phenotype but expression of CHOP alone is not sufficient to cause these pathologies since it lacks other interacting factors. This is evident from the interaction of CHOP with ATF4 in TM cells and this interaction does not occur when there is no ATF4. In addition, knockout of CHOP prevents ATF4-induced IOP elevation and TM cell death, further supporting this concept. We have modified our results as “ATF4-induced IOP elevation requires CHOP”. We have also included these points in the Discussion as “ It is likely that increased expression of CHOP in TM is involved in key glaucoma pathology but expression of CHOP alone is not sufficient to cause these pathologies since it lacks other interacting factors of chronic ER stress including ATF4”.

We have not conducted gene expression studies in this manuscript for two reasons. First, a previous study by Han et al, 2013 performed ChIP and mRNA-seq analysis and clearly demonstrated that ATF4 and CHOP target genes involved in protein synthesis. Second, it is likely that ATF4 may regulate several pathways and since our manuscript is focused on ATF4-induced protein synthesis and TM cell death, we thought these results may distract from our main focus. We feel that performing gene expression

studies is a great idea and future studies in the lab will be aimed at understanding the differential gene expression between ATF4 and CHOP in different cell types. As pointed out in the discussion, we hypothesize that ATF4 and CHOP may regulate differential gene expression in different cell types depending on interacting partners. For example, in our preliminary studies (part of another study), we observed that expression of ATF4 or CHOP alone can lead to neuronal cell death while only ATF4 (but not CHOP) leads to TM cell death. It would be interesting to compare gene expression between TM cells and neurons to better understand the roles of ATF4 and CHOP in cell death. In our revised manuscript, we have included analysis of several key pro- and anti-apoptotic markers. Surprisingly, we observed that ATF4 did not alter DR5 or other pro- or anti-apoptotic markers despite inducing cleaved caspase-3 and PARP. It is likely that ATF4-induced cell death is independent of DR5 in TM cells.

Comment 3) The data for the protein synthesis changes in Fig. 5A was not convincing. The model that induced ATF4 expression enhances protein synthesis and this is detrimental for TM and tissues was not convincing. This is the heart of the proposed model in the manuscript and one would need to incorporate additional markers and controls to sufficiently support the stated model. For example, more convincing protein synthesis measurements, measurements of total and phosphorylated eIF2 and GADD34, and consider an artificial block in translation (e.g. modest amounts of cycloheximide or the like) to address whether lowered translation restores health.

Authors Response: We thank the reviewer for providing these valuable suggestions. We have now revised Figure 5 and included these additional experiments. **1)** Western blot analysis clearly demonstrating that expression of ATF4 but not CHOP reduces phosphorylation of eIF2 α and concomitantly increased GADD34. Together, these data indicate that ATF4 promotes translational recovery (Fig. 5A). **2)** As suggested by the reviewer, we now show that treatment with mild doses of cyclohexamide (CHX) prevents ATF4-induced protein synthesis (Fig. 5B) and reduces TM cell death significantly (Fig. 6F-G). **3)** We also demonstrate that ATF4-induced protein synthesis and TM cell death is partly dependent on GADD34, as knockdown of GADD34 prevents ATF4-induced protein synthesis and TM cell death (Fig.6H-I). We strongly believe that these data strengthen our conclusions that ATF4 induces protein synthesis, which is associated with TM cell death.

Comment 4) The manuscript proposes that the reason why ATF4 is deleterious to TM is that the transcription factor (associated with CHOP or other partner) triggers increased translation (through GADD34?) and elevated ER client load (chronic UPR stress). The manuscript later indicates that ISRIB lowered Dex-treated IOP and TM fibrosis. ISRIB would be expected to thwart translational control by PERK induced eIF2-P and as a consequence block ATF4 and CHOP expression. However, ISRIB would also be expected to allow for high levels of mRNA translation during Dex-induced ER stress. If the model is that ATF4 is deleterious to cells because it allows for increased translation and overload of ER, would not ISRIB also drive high levels of protein synthesis during the Dex-induced ER stress? This would appear to be inconsistent with

the model. Perhaps ATF4/CHOP target genes involved in cell death pathologies (DR5 and the like) are critical to these pathologies?

Authors Response: We agree with the reviewer that ISRIB is expected to increase protein synthesis, which would be against our general hypothesis. We propose that protein synthesis in TM cells is tightly controlled by the ATF4-CHOP-GADD34 signaling axis and phosphorylation of eIF2 α . While ISRIB may transiently prevent phosphorylation of eIF2 α , which may result in a transient stop in translational brake, leading to increased protein synthesis, its inhibitory effects on ATF4-CHOP-GADD34 are more lasting and predominant in TM cells. The inhibitory effect of ISRIB on GADD34 would result in reduced phosphatase to inhibit phosphorylation of eIF2 α , which will further reduce protein synthesis. Alternatively, there may be another pathway by which GADD34 directly regulates protein synthesis. In support of this, our additional experiments clearly demonstrate that treatment with ISRIB significantly reduces Dex-induced protein synthesis in TM cells (contrary to published studies), which is associated with a strong reduction in GADD34, ATF4 and CHOP levels while its effects on phosphorylation of eIF2 α are minimal. Consistent with our findings, a recent study by Wang et al, 2019 independently demonstrated that the eIF2 α dephosphorylation inhibitor, Salubrinal, prevents tunicamycin-induced TM cell death. Moreover, dominant negative inhibitor of ATF4 reduced Dex-induced protein synthesis and ocular hypertension. It is also likely that ISRIB has differential effects in chronic ER stress compared to ISR since most of effects of ISRIB are studied in context of ISR pathway. In line with this, another study by Rabouw et al, 2019 demonstrated that ISRIB promotes protein synthesis when p-eIF2 α levels are low but ISRIB is ineffective when p-eIF2 α are high (in context of ISR). We have added this explanation in the Discussion.

Comment 5) The increase in cellular ROS due to overexpression in ATF4 is confusing. One often views ATF4 as having antioxidation properties (see Harding 2003 Mol Cell for lead-in). It is noted that ATF4 is tied to mitochondrial stress adaptation and resumption of translation can affect influx of proteins into the ER and disulfide bond formation, which can affect ROS.

Authors Response: We examined the role of ROS in ATF4-induced TM cell death because multiple studies have demonstrated the role of oxidative stress in ATF4-induced cell death (Han et al, 2013 and Lange et al, 2008). Atf4^{-/-} fibroblasts or neurons were resistant to oxidative stress (Harding et al, 2003 and Lange et al, 2008) and forced expression of ATF4 led to cell death which was reduced significantly by treatment with antioxidant (BHA) (Han et al, 2013 and Lange et al, 2008). Han et al, 2013 demonstrated that ATF4 and CHOP increase protein synthesis leading to oxidative stress and cell death. We therefore examined whether overexpression of ATF4 leads to oxidative stress and whether reduction of ATF4 prevents TM cell death. Contrary to studies by Han et al, 2013, reduction of oxidative stress did not prevent ATF4-induced cell death. There is no doubt that further study of the role of ROS and oxidative stress in glaucoma is warranted. Such studies are beyond the scope of the current study, but we report our current findings to indicate this fact, as is true for the role of ROS and oxidative stress in other diseases.

Reviewers' comments:

Reviewer #1 (Remarks to the Author):

Thank you for addressing the previous comment about the limited ATF4 overexpression in the glaucomatous human tissue. In response, the authors have added western blots and new immunofluorescence images to support their claim about ER stress in glaucoma. I must say, though, I am not blown away by the ATF4 blots. Fig 1A looks to me like ATF4 expression is very low in one of the normal TM lines, and maybe a bit elevated in the middle glaucoma lane, but 2 of the normals and 2 of the glaucomatous seem to have similar levels. This contrasts with the several-fold difference reported in the densitometry bar chart in panel B. A similar story with CHOP. The differences in GADD34 seem a bit more convincing, but there is lots of expression variability in the glaucomatous cells. Also, please clarify whether the "N=3" in the caption refers to technical replicates or different cell strains.

The IF image shown in panel C is borderline in terms of quality. Is this really your best set of images? It looks to me like the image is centred on anterior insert region of the TM, rather than the drainage portion, but it is difficult to tell. An SC endothelial label would help (CD31). There also seems to be an artefact (a thread?) in the upper panel giving some autofluorescence in the ATF4 channel.

Also, please acknowledge the potential that these markers may be upregulated in response to glaucoma therapies, which presumably the glaucomatous donors were receiving.

Thank you for addressing the previous concern about ER stress being sensitive to protein load and promoters. Just some clarification, the data shown in SI4 refers to Ad5.GFP, but the methods say only lentiviral GFP. Also, please clarify the pfu/TU (as appropriate) for the GFP in the caption.

There has been a noticeable pivot away from fibrosis as a mechanism for outflow dysfunction. This is good, but what is the explanation for why outflow is impaired in response to ER stress? Resistance is a physical property, and thus there must be some physical change in the outflow pathway. The authors should consider whether ER stress has an effect on TM or SC cell stiffness. I heard that there are a few engineer types in the glaucoma outflow field who might be interested in such an idea.

Reviewer #2 (Remarks to the Author):

This revised manuscript addresses the pathological mechanisms for primary open-angle glaucoma (POAG). Using in vitro and in vivo models, the manuscript indicates that induction of ATF4 and inappropriate resumption of translation (via expression of GADD34) is central to the pathologies of POAG. The central conclusions are solid and consistent with UPR mechanistic concepts in other organ pathologies. The study would be of broad interest. Prior concerns were sufficiently addressed.

It is noted that the authors' response to prior concern #4 was a bit muddled. The response stated that "While ISRIB may transiently prevent phosphorylation of eIF2 α , which may result in a transient stop in translational brake, leading to increased protein synthesis, its inhibitory effects on ATF4-CHOP-GADD34 are more lasting and predominant in TM cells." This statement was in the response letter but not in the manuscript. ISRIB does not block eIF2 α phosphorylation in response to ER stress. In fact it would arguably increase eIF2 α phosphorylation. ISRIB in part thwarts the ability of eIF2 α phosphorylation to lower its ability to inhibit the eIF2B GEF. As a consequence, there is elevated eIF2-GTP and sustained general translation even during stress and as noted in the later portion of the authors' response sentence above, the ATF4/CHOP/GADD34 transcriptional and translational expression would be lowered.

Authors Response Letter

Dear Reviewers,

We greatly appreciate the reviewers' valuable insights and suggestions, which have substantially improved our manuscript. We have carefully addressed all concerns and performed additional experiments to address these concerns. These changes are highlighted in the revised manuscript. Detailed responses to the reviewers concerns are listed below:

Reviewer#1:

1) Thank you for addressing the previous comment about the limited ATF4 overexpression in the glaucomatous human tissue. In response, the authors have added western blots and new immunofluorescence images to support their claim about ER stress in glaucoma. I must say, though, I am not blown away by the ATF4 blots. Fig 1A looks to me like ATF4 expression is very low in one of the normal TM lines, and maybe a bit elevated in the middle glaucoma lane, but 2 of the normals and 2 of the glaucomatous seem to have similar levels. This contrasts with the several-fold difference reported in the densitometry bar chart in panel B. A similar story with CHOP. The differences in GADD34 seem a bit more convincing, but there is lots of expression variability in the glaucomatous cells. Also, please clarify whether the "N=3" in the caption refers to technical replicates or different cell strains.

Authors Response: We very much appreciate the reviewer for providing insightful suggestions. Regarding Fig.1A, we have utilized 3 to 6 different cell strains of primary human TM cells (N=3 cell strains for ATF4 and CHOP; N= 6 cell strains for GADD34; N=4 cell strains for XBP-1). We have now included this information in Figure legends. Please note that densitometric analysis shown in Fig. 1B is an average of 2 independent replication of Western blots and the relative levels of ATF4 or CHOP is calculated over GAPDH (as a loading control). Therefore, relative levels of ATF4 or CHOP may be slightly different that representative blots. This variability in expression is commonly observed in human donor cells or tissues likely due to effects of IOP lowering drugs and genetic variability. Despite these variations, it is noteworthy that overall pattern of chronic ER stress pathway is activated in glaucomatous TM cells and tissues in cell culture, mouse models and human donor eyes as summarized below. First, immunohistochemical analysis of normal and glaucomatous TM tissues from post-mortem human donors, which clearly demonstrates significantly increased ATF4 and GADD34 in glaucomatous TM tissues (Fig.1C). Second, ATF4, CHOP and GADD34 were significantly increased in TM tissues of mouse model of Dex-induced ocular hypertension (Fig.1F-G and Supplementary Fig. 2). Third, we have previously shown significantly increased ATF4 and CHOP in different set of glaucomatous human primary TM cells (n=4 normal and 4 glaucomatous primary TM cell strains; Peters et al, 2015). We have also shown that glaucoma insults such as Dex (Zode et al, 2014) or mutant myocilin (Zode et al, 2012) induce chronic ER stress markers in mouse TM and in human primary TM cells.

2) The IF image shown in panel C is borderline in terms of quality. Is this really your best set of images? It looks to me like the image is centered on anterior insert region of the TM, rather than the drainage portion, but it is difficult to tell. An SC endothelial label would help (CD31). There also seems to be an artefact (a thread?) in the upper panel giving some autofluorescence in the ATF4 channel.

Authors Response: We have now repeated immunostaining for ATF4 and GADD34 (Fig.1C) to improve the quality of image. We have also included lower (Supplementary Fig. 1) and higher (Fig. 1C) magnification images to further focus on drainage structures including SC. Our brightfield images in revised Fig.1C clearly shows both TM and SC region and these regions are clearly marked. Therefore, we did not feel the need to stain SC with CD31. Increased ATF4 and GADD34 staining can be clearly seen in glaucomatous TM tissues but not in SC. We have also clarified that artifact (shown by *) is a fold in the tissue that can occur sometimes during sectioning of paraffin-embedded tissues.

3) Also, please acknowledge the potential that these markers may be upregulated in response to glaucoma therapies, which presumably the glaucomatous donors were receiving.

Authors Response: We have now included the following statements in the Discussion to address the potential effects of glaucoma therapies or other factors on ER stress.

“In the present study, we have shown that ATF4, CHOP and GADD34 levels are increased in human primary TM cells and tissues from POAG donor eyes. In addition, we have previously shown that CHOP is increased in human glaucomatous TM tissues. Of note, several of glaucoma donors are likely to be treated with IOP lowering drugs, which may positively or negatively alter ER stress markers. Since similar induction of ATF4, CHOP and GADD34 was also associated with IOP elevation in mouse model of Dex-induced ocular hypertension and in human primary TM cells from glaucoma donors, it is likely that induction of chronic ER stress is truly a key feature of glaucomatous TM pathology.”

4) Thank you for addressing the previous concern about ER stress being sensitive to protein load and promoters. Just some clarification, the data shown in SI4 refers to Ad5.GFP, but the methods say only lentiviral GFP. Also, please clarify the pfu/TU (as appropriate) for the GFP in the caption.

Authors Response: We have now mentioned that adenoviral and lentiviral vectors expressing GFP were utilized in Fig. 3A and SI.5 respectively. We have also clarified pfu or TU/eye in the Methods and Figure legends.

5) There has been a noticeable pivot away from fibrosis as a mechanism for outflow dysfunction. This is good, but what is the explanation for why outflow is impaired in response to ER stress? Resistance is a physical property, and thus there must be some physical change in the outflow pathway. The authors should consider whether ER stress

has an effect on TM or SC cell stiffness. I heard that there are a few engineer types in the glaucoma outflow field who might be interested in such an idea.

Authors Response: Our current and previous studies demonstrate that chronic ER stress leads to TM dysfunction and cell loss, which is likely to be responsible for increased outflow resistance. Loss of TM cells is associated with POAG (Alvarado et al., 1984). It has long been speculated that a loss of TM cells would lead to denudation of collagen structures and/or resulting in the collapse of the beams within the TM tissue. Such a collapse in the beams would decrease intertrabecular spaces thus restricting aqueous outflow and increase in IOP. Subsequently, these structural changes are likely to alter TM mechanics although this effect would be a secondary effect and not the direct causation of IOP increase. Based on the reviewer's suggestion, we examined whether ATF4 increases selected factors that determine TM stiffness including changes in actin cytoskeleton, rho-associated coiled-coil-containing protein kinase 1 (ROCK) and phosphorylation of myosin light chain 2 (p-MLC). We specifically focused on TM stiffness because we observed increased ATF4 and GADD34 only in TM but not in SC cells (Fig.1C). Indeed, we observed that expression of ATF4 significantly modulates proteins that regulate cytoskeletal remodeling which may in turn influence TM cell stiffness and contractility in human TM and mouse TM (S. Fig.19-20). Specifically, ATF4 increased alpha smooth muscle actin (S.Fig. 19) and also induced phosphorylation of MLC2 (S.Fig.20) in mouse TM tissues; these proteins are implicated in cell contractility. ATF4 also induced cleaved ROCK1 (an important substrate for caspase-3 to dictate apoptosis; Chang et al., 2006) and phosphorylation of MLC2 in human TM cells (S.Fig.20). These data demonstrate that ATF4-mediated dysfunction in TM cytoskeleton and factors that regulate contractility is associated with TM cell death and IOP elevation. We strongly feel that understanding the effects of chronic ER stress on TM biomechanics warrant a separate investigation and our future studies will be aimed at performing meticulous ultrastructural, compositional, morphological, biochemical, and biomechanical characterization of the TM ECM with respect to chronic ER stress.

Reviewer#2:

This revised manuscript addresses the pathological mechanisms for primary open-angle glaucoma (POAG). Using in vitro and in vivo models, the manuscript indicates that induction of ATF4 and inappropriate resumption of translation (via expression of GADD34) is central to the pathologies of POAG. The central conclusions are solid and consistent with UPR mechanistic concepts in other organ pathologies. The study would be of broad interest. Prior concerns were sufficiently addressed. It is noted that the authors' response to prior concern #4 was a bit muddled. The response stated that "While ISRIB may transiently prevent phosphorylation of eIF2 α , which may result in a transient stop in translational brake, leading to increased protein synthesis, its inhibitory effects on ATF4-CHOP-GADD34 are more lasting and predominant in TM cells." This statement was in the response letter but not in the manuscript. ISRIB does not block eIF2 α phosphorylation in response to ER stress. In fact it would arguably increase eIF2 α phosphorylation. ISRIB in part thwarts the ability of eIF2 α phosphorylation to lower its ability to inhibit the eIF2B GEF. As a consequence, there is elevated eIF2-GTP and

sustained general translation even during stress and as noted in the later portion of the authors' response sentence above, the ATF4/CHOP/GADD34 transcriptional and translational expression would be lowered.

Authors Response: We greatly appreciate the reviewer's comment and we agree that ISRIB does not block eIF2 α phosphorylation in response to ER stress. We have now corrected this oversight from the manuscript and further clarified our Discussion (page 12) as

“ It is intriguing that ISRIB reduced Dex-induced protein synthesis despite several studies have shown that ISRIB increases protein synthesis by enhancing activity of eIF2B^{73,85,86}. Since ISRIB blocks downstream effects of phosphorylated eIF2 α , it can inhibit ATF4-CHOP-GADD34 signaling and also restore general protein synthesis. We propose that ISRIB's inhibitory effect on Dex-induced protein synthesis is primarily driven by its ability to effectively block ATF4-CHOP-GADD34 signaling in TM cells. ISRIB's inhibitory effect on GADD34 would result in reduced phosphatase to inhibit phosphorylation of eIF2 α , which will further reduce protein synthesis.”

REVIEWERS' COMMENTS:

Reviewer #1 (Remarks to the Author):

The authors have satisfied all previous concerns.